

# Exploring the opportunities and challenges of using large language models to represent institutional agency in land system modelling

Yongchao Zeng[1], Calum Brown [1,2], Joanna Raymond[1], Mohamed Byari[1], Ronja Hotz[1], Mark Rounsevell[1,3,4]

[1] Institute of Meteorology and Climate Research, Atmospheric Environmental Research (IMK-IFU), Karlsruhe Institute of Technology, 82467 Garmisch-Partenkirchen, Germany

[2] Highlands Rewilding Limited, The Old School House, Bunloit, Drumnadrochit IV63 6XG, UK

[3] Institute of Geography and Geo-ecology, Karlsruhe Institute of Technology, 76131 Karlsruhe, Germany

[4] School of Geosciences, University of Edinburgh, Drummond Street, Edinburgh EH8 9XP, UK

**Correspondence:** Yongchao Zeng (yongchao.zeng@kit.edu)

**Abstract** Public policy institutions play crucial roles in the land system, but modelling their policy-making
processes is challenging. Large Language Models (LLMs) offer a novel approach to simulating many
different types of human decision-making, including policy choices. This paper aims to investigate the
opportunities and challenges that LLMs bring to land system modelling by integrating LLM-powered
institutional agents within an agent-based, land use model. Four types of LLM agents are examined, all of
which, in the examples presented here, use taxes to steer meat production toward a target level. The LLM
agents provide reasoning and policy action output. The agents' performance is benchmarked against two
baseline scenarios: one without policy interventions and another implementing optimal policy actions
determined through a genetic algorithm. The findings show that while LLM agents perform better than the
non-intervention scenario, they fall short of the performance achieved by optimal policy actions. However,
LLM agents demonstrate behaviour and decision-making, marked by policy consistency and transparent
reasoning. This includes generating strategies such as incrementalism, delayed policy action, proactive
policy adjustments, and balancing multiple stakeholder interests. Agents equipped with experiential learning
capabilities excel in achieving policy objectives through progressive policy actions. The order in which
reasoning and proposed policy actions are output has a notable effect on the agents' performance, suggesting
that enforced reasoning guides as well as explains LLM decisions. The approach presented here points to
promising opportunities and significant challenges. The opportunities include, exploring naturalistic
institutional decision-making, handling massive institutional documents, and human-AI cooperation.
Challenges mainly lie in the scalability, interpretability, and reliability of LLMs.



## 1. Introduction

Land system models are increasingly incorporating elements of agency in land use and management decision-making. This process has several motivations, from theory-testing and exploration, to more predictive outputs based on process-based knowledge (Groeneveld et al., 2017). Such models can be particularly useful for understanding behavioural constraints on political strategies such as land-based climate mitigation (Perkins et al., 2023) or nature conservation through protected areas (Staccione et al., 2023). Agent-based land use models have now been applied from village to continental scales, revealing numerous ways in which land manager behaviour affects the rate, spread, and impacts of land use change (Brown et al., 2018; Kremmydas et al., 2018; Marvuglia et al., 2018; Matthews et al., 2007; Rounsevell et al., 2014).

Despite the growth in land use models based on agency, and despite their frequent application to policy questions, the nature and effects of agency among political and institutional actors have been relatively neglected. Institutions in general (spanning a wide range from informal social groupings to highly formal governance bodies) have almost exclusively been modelled as exogenous forces that alter model input settings in pre-defined ways, rather than as active participants in simulated land use change decision-making (Brown et al., 2017; Holman et al., 2019; Krawchenko and Tomaney, 2023). Meanwhile, evidence that institutions play key roles in land use change processes, and that these roles are strongly mediated by the agency of those institutions, has continued to grow (Dryzek, 2016; Dubash et al., 2022; Young et al., 2006). These institutions display a variety of key behaviours including inertia in decision-making, interaction among themselves, the use of partial or otherwise imperfect information, susceptibility to lobbying and social norms, and occasional abrupt changes in objectives. These types of processes pose a substantial challenge to representation in land system models.

The rise of Large Language Models (LLMs) provides a novel and potentially powerful approach to modelling the decisions of institutional agents. LLMs are a class of Artificial Intelligence (AI) models designed to understand and generate human-like language (Brown et al., 2020; Devlin et al., 2019; Vaswani et al., 2023). They have recently been applied to computational agent design bringing benefits for both fields (Sumers et al., 2023; Wang et al., 2023; Weng, 2023; Xi et al., 2023; Yao et al., 2023). LLM-powered agents are by the nature of their design and training implicit models of human decision-making and simulations using language agents can produce believable human behaviour in various contexts (Horton, 2023; Park et al., 2023). They form opinions, interact with one another and with the user in natural language, learn from experience and make plans for the future in ways that are similar to humans. This makes LLMs a powerful tool for modelling the decision-making and behaviour of institutional agents which interact dynamically with their environment.



Effective LLM-powered agents are pre-trained using massive amounts of textual data containing diverse linguistic patterns. In principle, therefore, LLM agents can consider a wider range of factors, transcend the paradigm of economic rationality and generate more nuanced, context-aware and adaptive responses to specific problems. In contrast to traditional agents, they can generate novel or unexpected behaviour supported by explicit reasoning, which provides an opportunity to search for novel insights into human behaviour in the real world. Conversely, if LLMs are used without sufficient understanding or interpretation, they can act as amplifiers of biased or erroneous data, uninformative 'black box' models or distractions from more useful approaches.

In this paper, we explore a novel application of LLMs to represent the behaviours of public policy institutional agents in a large-scale, agent-based model of the land system. We seek to represent the decision processes of policy agents through LLM simulations that are constructed through the support of a human operator. We design a set of LLM-powered institutional agents and couple them with the CRAFTY land use model (Murray-Rust et al., 2014). CRAFTY serves as an uncertain, dynamic environment where institutional agents use limited information to achieve a well-defined policy goal by employing strategic policy actions that influence land users' decision-making. The institutional agents' performance and behavioural patterns are evaluated and analysed and the reasoning behind a sequence of selected policy actions is investigated in detail. The overall purpose is to explore the opportunities and challenges of LLM in modelling policy institutions beyond existing (albeit limited) approaches.

## 2. Methodology

### 2.1 Human-operator-centred prompt development

In contrast to conventional approaches that hard-code agents' behaviours, an LLM-powered agent operates based on prompts given in natural language. The efficacy of an LLM in a simulation hinges critically on the quality of the LLM and the prompts employed. The quality of an LLM itself is largely dependent on the LLM's providers. LLM end-users mainly leverage prompts to communicate with and instruct the LLM to achieve specific goals. Although a prompt is simply a user input that an LLM is expected to respond to, creating an effective prompt template is an intricate process, particularly when integrating LLM-powered agents into specialized simulation environments. Our methodology for prompt development encompasses a four-stage process: Discovery, Drafting, Fake-Loop Testing, and Real-Loop Testing, in which the LLM is supported by continuous engagement and refinement by a human operator.

a. **Discovery:** Prompt engineering is a rapidly developing area and a wide range of useful prompt templates have now been developed and published for various purposes**.** The initial phase is dedicated to comprehensive research, including reviewing relevant literature and online searches for existing templates that might align with the simulation needs. Owing to the unique aspects of the simulation



model presented here, finding a fully formed template was not possible. However, valuable insights and components can often be gleaned during this phase. For instance, few-shot learning (Brown et al., 2020) and chain-of-thought (Wei et al., 2022) are both useful and generalizable prompt techniques that can serve a variety of purposes.

b.  **Drafting:** If a suitable pre-existing template cannot be found, the next step is to construct an initial draft.
This draft must clearly describe the tasks to be performed by the LLM. Utilizing ChatGPT as a drafting tool has the advantage of its extensive pre-training data that may encompass a broad range of prompting techniques and high-quality prompt templates. Nonetheless, the outputs generated by ChatGPT must undergo rigorous examination and iterative refinement by the human operator to ensure alignment with the simulation objectives.

c.  **Fake-Loop Testing:** Upon reaching a satisfactory draft, we proceed to the fake-loop test. This stage is particularly beneficial when running actual simulation models is resource-intensive. Here, simulated data—crafted by experts familiar with the simulation model—serve as a stand-in for simulation outcomes, allowing for assessment of a prompt without the need for running an actual simulation. This enables swift identification and rectification of issues within the prompt.

115   d.  **Real-Loop Testing:** Successful fake-loop testing paves the way for the real-loop test, which entails the integration of the LLM with the actual simulation model. However, challenges may arise, such as outputs that disrupt the simulation due to formatting errors, necessitating a restart. To mitigate such setbacks, a Human-in-Loop (HIL) approach is used during the real-loop testing phase to enhance the prompt template's robustness and reliability.

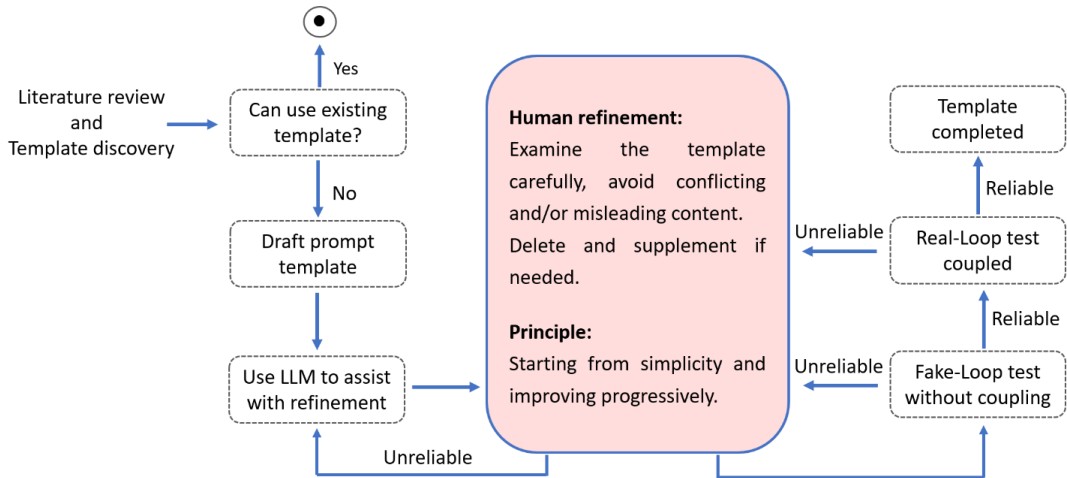

Figure 1    The operational flowchart of human-operator-centred prompt development



Through this structured approach, we refined the integration of LLM-powered agents within the simulation framework, ensuring that the prompt design was not only effective but also adaptable to the dynamic nature of real-world simulations.

## 2.2 Applying Human-in-Loop (HIL) design to a real simulation

Incorporating a HIL approach in the real-loop testing phase offers substantial benefits, enhancing interactivity and adaptability, which can lead to significant time and cost savings throughout the development process. As depicted in Fig. 2, the process commences with an initial prompt template as input to the LLM model. This template includes foundational information for the LLM and placeholders for dynamic updates. Upon processing this input, the LLM formulates a policy proposal.

At this juncture, a human operator is required to assess the LLM's output for its rationality and formatting. Should the output fall short of expectations (e.g., misunderstanding the tasks, illogical output or inaccurate formatting), the operator marks a Boolean variable as false, signifying the proposal's rejection. Accompanying this action, the operator provides feedback intended to refine subsequent responses from the LLM. For instance, the LLM agent may misunderstand its objective and propose actions that are not considered in the land use model. The operator can leave a comment to emphasize its objective and the boundary of action space it should focus on. This commentary, alongside the original LLM proposal, is woven into a dialogue that iteratively informs the prompt's evolution.

The dialogue between the LLM and the operator is preserved in the agent's "memory", ensuring that the LLM's learning is cumulative and contextually aware. The actionable part of the LLM's final, operator-confirmed proposal is then extracted and incorporated into the simulation model. This model represents the institutional environment within which the agent operates, and it generates outcomes based on the agent's actions. These outcomes, in turn, become part of the feedback loop, informing the agent's proposals in the subsequent iteration.

This HIL process is crucial for maintaining a dynamic and responsive testing environment, where human expertise plays a pivotal role in guiding the LLM to generate proposals that fit with the constraints of the task and the simulation to be coupled with. The HIL design can serve multiple objectives. Primarily, it leverages human examination to promptly identify and correct any issues with the LLM's responses. For instance, if the LLM misunderstands its instructions, a human operator can clarify the error via comments without halting the entire simulation. This capability is useful, especially in the initial stages of simulation when the prompt template may not be fully refined. It allows operators to observe a broader range of responses from the LLM, accumulating insights that are instrumental in subsequent prompt refinement.



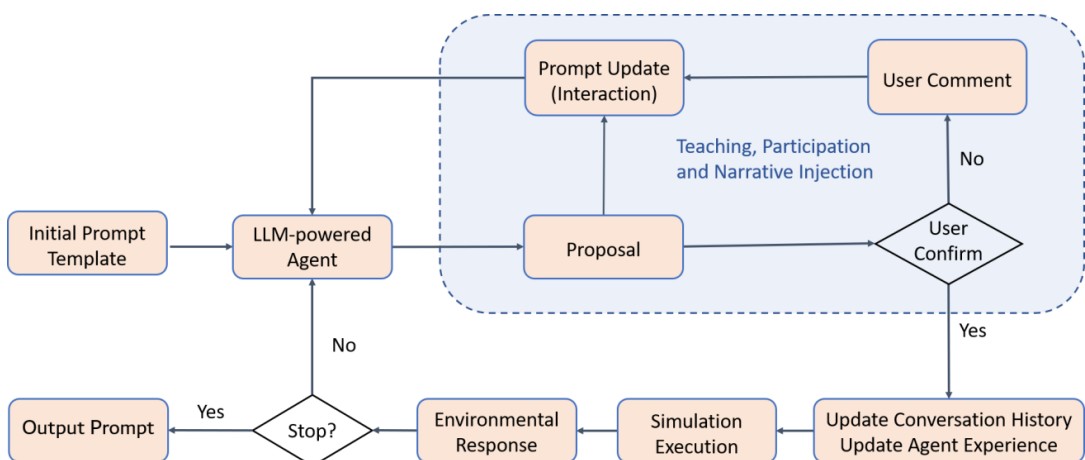

Figure 2 Human-in-loop (HIL) design applied to a running simulation model


A simple illustrative case is when the LLM generates a satisfactory proposal that fails to meet specific formatting requirements. An operator can guide the LLM by commenting, "Your proposal is plausible, but it needs to be formatted as follows...". Should the LLM continue to underperform after several interactions, the operator has the option to instruct the LLM to output a predefined result, bypassing a complete

simulation restart. This approach not only salvages the current simulation run but also garners additional data, enriching the prompt engineering process post-simulation.

## 2.3 Integration with the CRAFTY land use model

CRAFTY is an agent-based modelling framework designed for simulating large-scale, land use change (Blanco et al., 2017; Brown et al., 2018; Murray-Rust et al., 2014). The framework mimics land use

dynamics arising from the competition between, and strategic decisions of, different land users. The land users, represented by agents in CRAFTY, either individually or collectively, contribute to generating a diverse range of ecosystem services, utilizing various forms of natural capital, which represent the productive potential of the land and socio-economic capitals that represent the context within which agents make decisions. The land user agents within the model are categorized into discrete Agent Functional Types

(AFTs) (Arneth et al., 2014). This categorization is based on several criteria, including the intensity of land management and the characteristics of the agents' decision-making processes. Key factors in this categorization encompass the degree to which profit generation is prioritised and their tendency to conserve land. The basic model framework is described in Brown et al. (2018). This study uses a newly-developed emulator of the CRAFTY_EU application (Brown et al., 2019; Brown et al., 2021) that allows for rapid and

easily-adaptable simulations to be performed.



Here, the CRAFTY model is coupled with the LLM-powered-institutional agents that employ policy instruments to influence the land users' decisions on ecosystem service production. Fig. 3 illustrates the model processes encompassing the six steps that were implemented here:

1) CRAFTY was initialized by establishing the distribution of AFTs, capital maps, and demand parameters according to a specified Representative Concentration Pathway (RCP) (Van Vuuren et al., 2011) and Shared Socioeconomic Pathway (SSP) (O'neill et al., 2014) of climate change and socio-economic change scenarios, respectively.

2) The institutional agents were initialized by defining policy types and policy goals.

3) Data were collected from CRAFTY to capture the internal dynamics of the land use system.

4) Policies were adapted based on system observations, institutional evaluation and deciding on policy adjustments. If adaptation was necessary, the LLM agent suggested new policy actions. In the absence of adaptation, existing policies were maintained.

5) Policies were applied in the land use system (by changing the CRAFTY input for a specific iteration).

6) The objectives were evaluated by assessing whether the desired outcomes were achieved. If objectives were met, the process was concluded; if not, the cycle returned to Step 3 for further observation and adjustment.

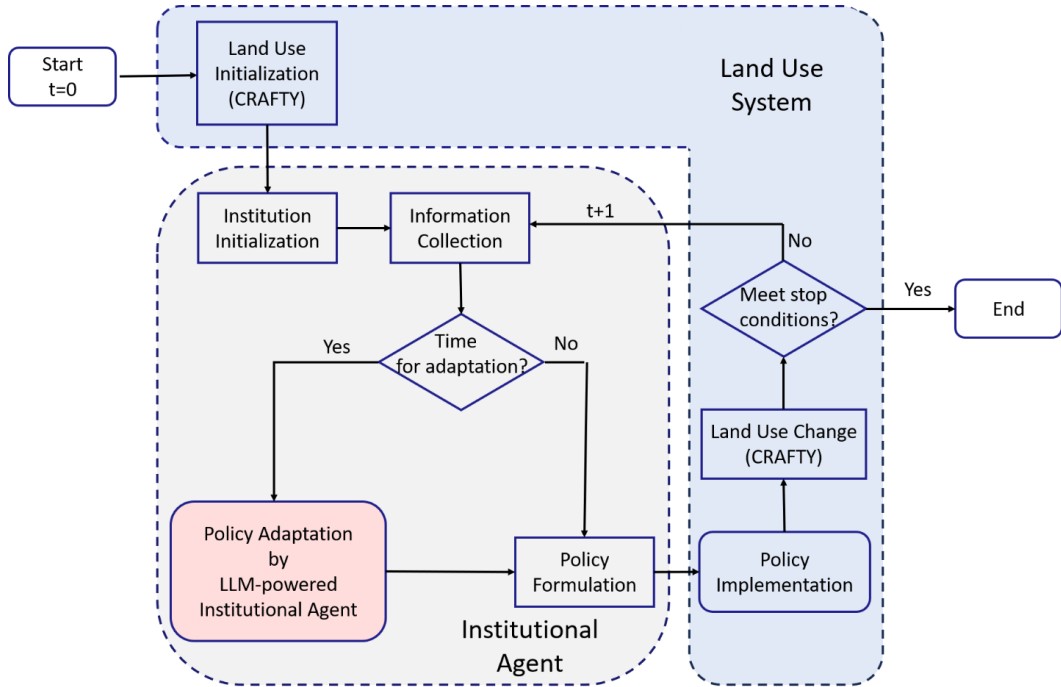

Figure 3 Coupling CRAFTY with LLM-powered institutional agents



## 2.4 Experimental settings

While CRAFTY considers a wide range of ecosystem products and services, the exploratory experiments presented here focused on a single ecosystem service, meat production, under the influence of institutional agents. Meat production has significant environmental impacts: it is a major contributor to deforestation, biodiversity loss and is the single most important global source of methane. Yet meat consumption continues to increase globally each year (Godfray et al., 2018), hence the plausibility of policy interventions. A

powerful economic incentive for changing consumption patterns is the implementation of meat taxes. Here, we assign the LLM-powered institutional agent the task of regulating meat supply through taxation, with the objective of aligning supply with a predetermined level. Although this task appears simple it presents significant challenges in terms of offsetting the impact of increasing demand for meat, dynamics with other connected ecosystem services, and the land use system not being fully known to the agent.

We designed six types of agents, including two non-LLM agent types, to conduct numerical experiments. The specifics of these agent types are given in Table 1. The prompts for the LLM-powered agent types are given in Appendix A. The LLM used here was gpt-4-1106-preview. All LLM agents were provided with two series of historical data for their decision-making: the gap between meat supply and the policy goal ('average errors'), and policy actions that were implemented. To mitigate linguistic confusion, the policy

actions are simplified into a finite space of eleven tax levels, represented by integers ranging from -5 to 5. The relevant equations and computations are given in Appendix B.

Table 1   Agent types included within the experiments and their corresponding features

| Agent types | Key features | Description |
|:---:|:---|:---|
| B1 | • Baseline agent;<br>• Not powered by LLM;<br>• Does nothing. | The role of Agent B1 in the simulation is equivalent to the absence of an institutional agent, mirroring the baseline scenario without any policy intervention. |
| B2 | • Baseline agent;<br>• Not powered by LLM;<br>• Policy optimizer. | Compared with B1, B2 is another extreme. B2 conducts a sequence of policy actions derived from a genetic algorithm that seeks optimal actions. |
| S1.1 | • Single agent;<br>• Powered by LLM;<br>• Outputting reasoning prior to final policy actions;<br>• No experiential learning. | S1.1 makes decisions based on the historical data provided but with no experiential learning, to ensure that reasoning is clear and non-iterative, and therefore easy to interpret. |
| S1.2 | • Single agent;<br>• Powered by LLM;<br>• Outputting final policy actions prior to reasoning;<br>• No experiential learning. | S1.2 operates as S1.1 with the exception of the order in which its actions and reasoning occur. This variation is investigated because the output sequencing is found to significantly impact the institutional agent's performance. |





| S2 | • Single agent; • Powered by LLM; • Output reasoning prior to final policy actions; • Using experiential learning to enhance decisions. | S2 should mimic human decisions more accurately than S1.1 and S1.2 as it uses experiential learning. This means that the agent produces substantially more textual output to explain its decision-making. |
|---|---|---|
| Q | • Quasi-multi-agents with five roles involved in decision-making; • Powered by LLM; • The five roles include policy analyst, government official, economist, meat producer representative, and environmentalist; • Output a conversation among five roles prior to policy actions; • No experiential learning. | Unlike traditional multi-agent systems where each role is modelled as a separate entity, quasi-multi-agents employ LLMs to simulate a cohesive dialogue among these roles. This methodology avoids the difficulty in sequencing the roles, and conversation endpoint setting, but saves time and token cost. |

To better illustrate the performance and behavioural patterns of the LLM-powered institutional agents, we
used Agent B1 and B2 to set up two baseline scenarios. The first baseline scenario reflects the simulation
without policy interventions; while the second used a genetic algorithm to seek optimal policy interventions
in which meat supply follows the prescribed target supply level. The genetic algorithm searches for a
sequence of policy actions that minimize the sum of squared average errors (gaps between meat supply and
the target level) across all the iterations in a simulation. These two baseline scenarios therefore give idealized
limits within which subsequent simulations can be situated.

## 3. Result analysis

### 3.1 Baseline scenarios

Figure 4 depicts meat demand (red arrowed line) and supply (blue solid curve) without policy intervention.
Initially, the meat supply mirrors the rising demand, exhibiting only minor fluctuations. The data spans 71
years, from year 0 to year 70, with additional simulation years extending beyond this period using the same
input data as in the 70th year. This extension allows us to observe the agent's performance in a relatively
stable environment without being influenced by the evolving meat demand. The policy goal, depicted as a
dashed horizontal line, is to maintain meat supply at its initial level, challenging the agent to use taxation as
a tool to minimize the discrepancy between actual output and this target. In reality, both policy objectives
and market demands are crucial for balanced policy-making. However, for this experiment, the policy goal
was intentionally set at an unrealistic level to exert additional pressure on the agent. In contrast to the
scenario without policy intervention, the green solid curve represents the resultant meat supply under the
optimal policy interventions.



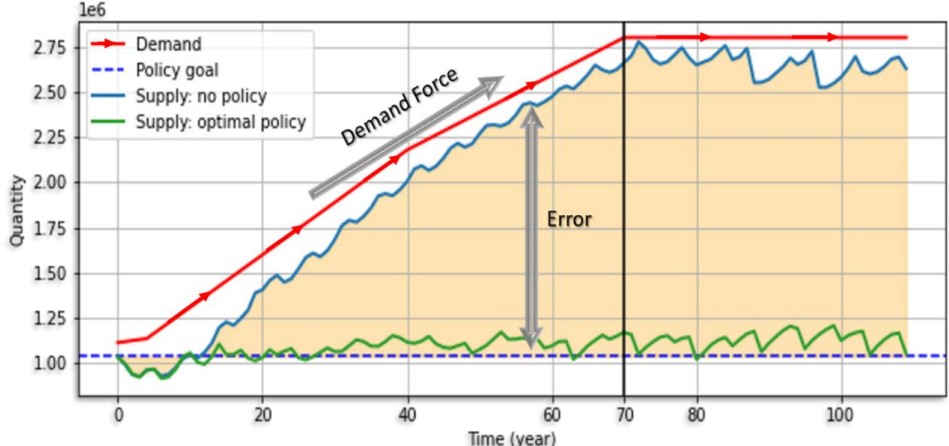

Figure 4   Changes in meat demand and supply without policy intervention. The policy goal (dashed horizontal line) is to maintain constant meat production. Fluctuations in supply are due to a lack of simulated behaviour affecting individual land manager agents' responses.

To visualize the agents' behaviours and corresponding outcomes, we use plots with dual vertical axes that simultaneously reflect the variation in the policy actions and in the average errors in the two baseline scenarios:

*Baseline Scenario 1: Agent B1 With No Policy Intervention.* This scenario is depicted in Fig. 5a. It shows the average error in meat output relative to the policy goals (left axis) and the absence of policy actions taken by the institutional agent (right axis). The average error trend reveals an increasing divergence from the policy goals, peaking at around the 70th year. After this period, the error rate stabilizes, reflecting a system in its steady state without further input updates.

*Baseline Scenario 2: Agent B2 With Optimal Policy Actions.* Contrasting the first, the second scenario, shown in Fig. 5b, adopts an approach based on optimization. Here, the policy actions vary significantly over time, representing dramatic annual changes that are unlikely to represent real-world policy-making. However, the curve representing the average errors exhibits an evident tendency to closely follow the horizontal axis, indicating the efficacy of these policy actions.

## 3.2 Performance of the LLM agent types

### 3.2.1 Performance of Agent S1.1

Figure 5c shows the performance of institutional Agent S1.1. Compared with Agent B1 without policy intervention, S1.1 has a notable impact on meat supply. The average error peaks between -120% and -100%



in the 70th year, in contrast with approximately -140% for the baseline scenario without policy intervention. A significant difference occurs after the 70th year. The average error approaches zero steadily, indicating that institutional Agent S1.1 has at that point in time found effective policy actions to achieve the policy goal. The policy actions taken by S1.1 are generally understandable. Initially, the meat supply is slightly below the policy goal, resulting in a positive average error. S1.1 chose to incrementally decrease the tax. When meat supply increases (driven by increasing demand, which the institutional agents are unaware of), S1.1 started to maintain or increase the tax (in contrast to the optimizing Agent B2, which chose policy actions that fluctuated irregularly). Starting from the fourth policy action, all the following policy actions are non-negative, suggesting the agent might be making plausible moves.

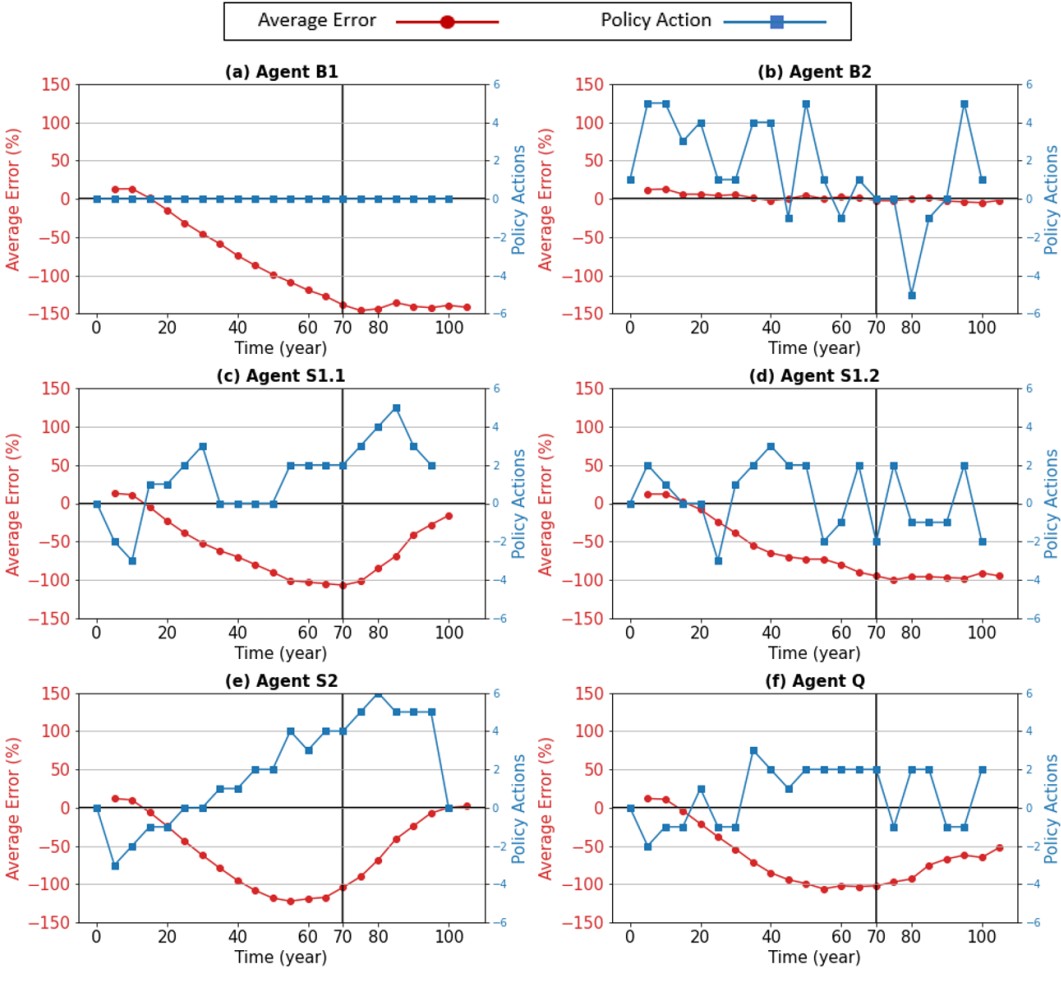

Figure 5 The average error in meat output relative to policy goals and the policy actions taken by the institutional agents



### 3.2.2 Performance of Agent S1.2

Figure 5d shows the performance of Agent S1.2. As described in Table 1, S1.2 uses the same prompt template as S1.1 but with a small difference in the order of the required output. S1.2 is required to output the final policy action before giving the rationale behind its decision. As can be seen, the policy actions taken by S1.2 are much less consistent. S1.2 also performs poorly after the 70th year and is unable to navigate meat supply towards the objective. We can see the reasoning behind its policy actions, using the second policy action as an example, which is to increase the tax by two levels when the average error is positive. As shown in Fig. C1 in Appendix C, it stated in the first sentence of its output that "A moderate tax decrease could be one approach", which is a reasonable action to mitigate the current under-supply issue. However, in the next paragraph, it contradicts this by proposing "+2" for the policy action, indicating an increase in tax. This decision was supported by complex reasoning: increasing tax can filter out inefficient meat producers and spur meat production technologies, which are better for the long-term sustainability goal. In comparison with the output of S1.1, S1.2 seems to provide reasoning in hindsight to justify a decision made in the absence of such reasoning. Another crucial issue captured in the output text of S1.2 is that some policy actions are given without a follow-up reasoning. Additionally, the required output format is often not strictly followed.

### 3.2.3 Performance of Agent S2

When contrasted with S1.1, S2 exhibits a notably incremental approach to policy actions, as shown in Fig. 5e. The tax level adjustments are mainly minimal, consistent with the smallest possible change. This pattern of incremental change is initiated from the second policy action and progressively escalates, reaching a higher intensity towards the simulation's end. Intriguingly, the policy action sharply reverts to zero in the final phase, suggesting that S2 reaches a decision to maintain the current tax level, deeming it optimal. This gradual and deliberate strategy in policy action results in a smoother meat supply curve, effectively meeting the set policy goal. Such measured and incremental actions align more closely with human decision-making processes, reflecting the nuanced impact of experiential learning in the scenario.

### 3.2.4 Performance of Agent Q

Agent Q epitomizes a quasi-multi-agent ensemble, embodying five distinct roles engaged in deliberation and negotiation (as shown in Table 1). Despite a concerted effort, the average error curve (see Fig. 5f) indicates that the group's performance was suboptimal. While the error magnitude was less severe than that of S1.2, it exceeded that of S1.1 and S2.

Upon examining the internal dialogues of Agent Q (Table A5 in Appendix A), the sophistication of the LLM becomes apparent. Each role upholds unique priorities and responsibilities, contributing to a multifaceted discussion. The discourse typically begins with the policy analyst, who accurately interprets





the data and highlights the supply shortfall relative to demand, reiterating the objective to sustain meat production at baseline levels. The government official then synthesizes insights from the collective, while the economist briefly evaluates the fiscal implications of tax adjustments. The meat producer representative and environmentalist voice their sector-specific concerns and policy preferences. Ultimately, the government official is tasked with formulating a policy response.

Although Agent Q's roles do not collectively achieve the policy goal, they offer an array of believable stakeholder perspectives—an indispensable aspect that poses a considerable challenge for conventional modelling approaches. The resulting policy actions reflect the inherent difficulty in harmonizing diverse interests. Notably, the government official's actions are characterized by prudence, as evidenced by the narrow range of policy adjustments, oscillating between -2 and +2, to avoid excessive opposition. This conservative approach underscores the complexity of policy-making in a multi-stakeholder context where a balancing act is as critical as the policy decisions themselves.

## 3.3 Dive into the Brain

While LLM models are often perceived as opaque, LLM-powered agents can offer the compelling ability to articulate human-comprehensible reasoning for their actions, providing a window into the decision-making processes that drive their behaviour. Such transparency is not only instrumental in validating the agents' credibility but also serves as a source of inspiration for enhancing institutional models and informing real-world policy decisions. One of the challenges, however, lies in the voluminous textual output generated when these agents are integrated with simulation models—making it impractical to display and analyse systematically.

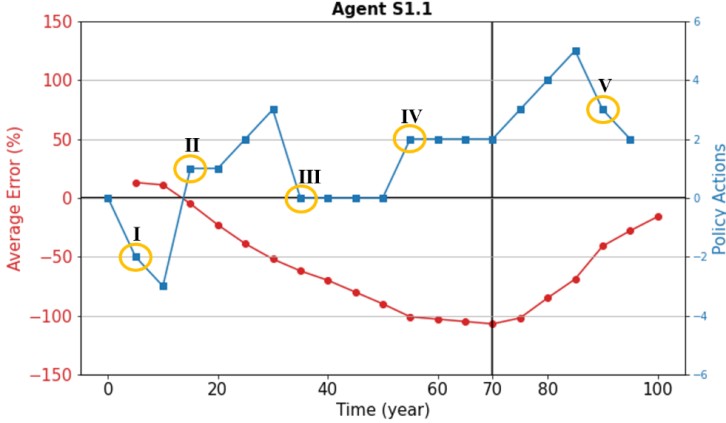

Figure 6    Selected key policy actions executed by Agent S1.1.



To navigate this, we concentrate on a subset of the data that offers significant insights. Specifically, we have distilled the textual output down to five key policy actions executed by Agent S1.1, focusing on the rationale
that underpins these decisions. Agent S1.1 is selected here because, in general, it demonstrated believable actions yet its reasoning is less history-dependent, which makes it easier to interpret through an in-depth investigation the large volume of textual output. This investigation provides a valuable glimpse into the "thought processes" of the institutional agent. Fig. 6 marks these pivotal moments, numbered using Roman numerals from I through V, allowing us to dissect and understand the logic applied at each juncture.

*Action I: How did the agent reason with insufficient information?*

The initial policy decision by an institutional agent is often the most challenging due to the lack of historical data. Detailed in Fig. C2 in Appendix C, the agent begins its reasoning by acknowledging this. The agent then turns to foundational economic principles to guide its decision-making process, aligning with the policy goal. The agent outlines the economic theory underlying the use of taxation to influence meat production
levels before delving into the specifics of the policy action required. It considers the industry's response time to policy changes and the potential for overreaction. After weighing these factors, the agent chose a conservative approach, adjusting the tax by a moderate "-2". This decision reflects a strategic balancing act: it is cautious to mitigate the risk of radical industry reactions, yet still steers towards the policy goal.

   *Action II: How did the agent deal with the first overshoot?*

Following a period of increased taxation, the institutional agent observed an overshoot in meat supply relative to the policy target. The agent conducted an analysis to identify the cause of this discrepancy. It concluded that the overshoot was a result of its earlier decision to reduce the tax by three levels. From the modeller's perspective, it is apparent that the primary driver of the overshoot was the rapidly increasing demand, rather than the tax adjustment, but the agent is not aware of this fact. Given the limited data
available to the agent, its attribution, while inaccurate, is understandable.

As shown in Fig. C3 in Appendix C, in response to this perceived causation, the agent selected a conservative corrective measure, implementing a modest tax increase of "+1" to rectify the minor discrepancy. Interestingly, the agent seemingly confuses the policy goal of maintaining supply levels with the objective of matching supply to demand. This confusion likely stems from the stochasticity inherent in
its generative response and the data with which it was provided. During the development of the prompts, we observed the agent's recurring misunderstanding of the objectives. To prevent further confusion and to streamline the agent's decision-making, we intentionally omitted demand information from the prompts. This decision highlights the challenge in prompt engineering, where the inclusion or exclusion of specific data points can significantly influence the agent's understanding and subsequent actions.





*Action III: How did the agent explain this counter-intuitive action?*

The third highlighted decision point presents a somewhat counterintuitive choice by the institutional agent, especially in the context of the rapidly expanding average error. Logic would suggest that the agent should further increase the tax to mitigate the excess in meat supply overshooting the policy target. However, as detailed in Fig. C4 in Appendix C, the agent opted to maintain the current tax level. This decision was based on its assessment that the market required more time to fully respond to its previous significant policy action of a "+3" tax increase.

This approach reflects the agent's consideration of the time lag inherent in market reactions to policy changes. The decision to hold steady on the tax rate, rather than implement further increases, was informed by the understanding that the "+3" adjustment was the most substantial move it had made since the simulation's inception. The agent's choice to allow the market time to adjust to this major policy shift, rather than immediately introducing another change, indicates a level of strategic foresight.

*Action IV: What led the agent to change its action?*

After a period of maintaining a consistent tax level, the institutional agent made a notable change, increasing the tax by two levels. As detailed in Fig. C5 in Appendix C, this decision appears to stem from the agent's growing suspicion that factors beyond the scope of its existing data and prompt instructions were influencing the market dynamics. Although this suspicion is speculative, it is a plausible consideration given the complexity of the land system it is dealing with, and was actually correct in this case.

However, the agent's analysis reveals a misinterpretation of the cause-and-effect relationship in the data. It mistakenly attributed the increasing average error to its prior decisions to raise taxes. While the data shows a negative correlation between tax increases and the average error, it is illogical to speculate that the tax hikes are solely responsible for exacerbating the situation. This misattribution stands in contradiction to the agent's subsequent decision to further increase the tax.

Moreover, by examining the agent's reasoning, one can notice the rationale provided by this agent is muddled. While the decision to increase the tax could be seen as a logical response to the perceived need for corrective action, the reasoning process the agent employs lacks clear logical coherence. This disconnect between the agent's final decision and its reasoning highlights potential areas for improvement in the agent's decision-making framework and the prompts that guide it.

*Action V: What made the agent brake?*

Action V represents a proactive approach. Upon reviewing the outcomes of its recent actions and the corresponding fluctuations in the average error, the agent acknowledged the effectiveness of these measures.



Recognizing the potential risks associated with overcorrection, especially given that its most recent policy involved the maximum possible increase in tax, the agent proceeded with caution.

In its decision-making process, as outlined in Fig. C6 in Appendix C, the agent carefully weighed the implications of further tax adjustments, comparing the potential outcomes of increasing the tax by +2, +3, and +4. Eventually, it settled on a +3 increase, which maintains the increasing trend of tax but at a slower pace, akin to a driver slowing down when the destination is close.

This reasoned and well-articulated approach in Action V notably contrasts with the less coherent rationale observed in Action IV. This disparity in the quality of reasoning between Action IV and Action V implies a key characteristic of LLM-powered agents: their performance can be variable and somewhat unpredictable. While Action V reflects a higher level of analytical sophistication and logical consistency, the inconsistency in performance across different iterations highlights the challenges in achieving stable and reliable outputs from LLM-powered agents. This variability points to the ongoing need for refinement and development in the application of LLMs in complex decision-making contexts.

## 4. Discussion

The experiments presented here reveal that LLM-powered agents, representing institutional decision-makers, display a spectrum of behaviours and reasoning processes that closely resemble human decision-making. These behaviours emerge naturally, unscripted by modellers, and encapsulate complex aspects of human cognition, which are traditionally challenging to simulate. At the same time, inconsistencies in decision-making within and between agents suggest specific challenges (as well as solutions) for the future use of LLM agents in this domain.

In our experiments, LLM-powered institutional agents are able to move modelled outcomes towards their objectives, but do so less well than an agent powered by an optimizing genetic algorithm. These results align with our expectations, especially given the bounded rationality and imperfect information available to the LLM. Among agents, we find that the ability to learn from past experience improves outcomes, as does, unexpectedly, a requirement to provide reasoning before making a decision. When this order is reversed, actions are found to be inconsistent with the post-hoc reasoning provided, suggesting that decisions in this case are made according to opaque internal mechanisms that the LLM does not explain in its spurious justifications.

In this study, we used GPT-4, which is a generative language model. Generative language models can be simply deemed as textual completion machines; they require prompts to initialize the context guiding their textual output, and newly generated texts add to the context for further output. That is, a generative language model uses its output to continuously update its context (Goldstein et al., 2022). Therefore, the order of



output does matter. One can confirm this by asking ChatGPT-4 a simple question: "Is 3 75% of 4?" This

question elicits an incorrect answer, followed by an admission of confusion and a correction (Fig. C7 in

Appendix C). If asked to give reasoning ("Is 3 75% of 4? Give your reasoning before answering"), the

response is correct (Fig. C8). This finding is consistent with the idea of chain-of-thought, which contends

that a generative language model performs better if it outputs answers step by step (Wei et al., 2022). The

step-wise output not only represents the outcomes but also a way of context building. A prompt might only

work as an initial trigger, but the generative language model needs self-prompting to complete the response

appropriately.

Although the textual completion functionality seems artificial, it is intuitively consistent with how humans

behave. It is normal for a human to have an illusion of understanding an issue until being required to

articulate or explain that issue to others, or to recognize logical gaps during verbal explanations (Ericsson

and Simon, 1998; Keil, 2006). In other words, we need to properly prompt our neurons to give appropriate

output. This does not imply anything more than superficial similarity in the behaviours of people and LLMs

(Fokas, 2023) – and this superficial similarity can easily mislead – but it does add interest to the use of

LLMs as agents in simulation models.

Further interest is provided by our experiment with the multi-faceted 'Agent Q'. While performing less well

than others in achieving its policy goal, this agent generated contextually coherent conversations between

five critical policy-relevant roles. The conversation captures each role's characteristics and interests,

particularly demonstrating the policymaker struggling to balance the interests of the meat producer and the

environmentalist. Agent Q is a group of quasi-multi-agents. Quasi here means the agents are different from

real multi-agents each of which has an independent and relatively complete cognitive system. Several

studies have applied LLM to multi-agent systems, where each agent has an independent cognitive system.

For instance, Park et al. (2023) built an artificial village consisting of 25 LLM-powered villager agents;

Qian et al. (2023) simulated a software development team with different roles. Agents in such multi-agent

systems have different personalities, targets, memories, etc., which together form a unique prompt triggering

their responses. Such systems are convenient for LLMs to generate short conversions between a pair of

agents, but can become cumbersome for conversations involving more than two agents. As above, the order

of text generation can affect performance in an LLM, and numerous equally-valid orders are possible

communicating in a group conversation, possibly leading to open-ended outputs. Our use of quasi-multi-

agents hands control to the LLM, saving time and token fees.

Besides investigating the quantitative performance of the agents, we also qualitatively analysed output of

Agent S1.1, which made decisions after providing reasoning and without learning from experience. This

analysis is a unique opportunity that conventional methods cannot provide. Agent S1.1 was found to eschew



drastic changes, instead opting for a series of cautious, incremental steps aligned with the principles of incrementalism — a well-known theory in political science (Pal, 2011), which posits that policymakers often employ heuristics and make modest, tentative changes to gradually achieve policy objectives. In addition, Agent S1.1 exhibits an acute awareness of the time lags inherent in the land use system's response to policy shifts. It strategically maintains a consistent tax rate, allowing time for the system to adapt and provide feedback—a practice mirroring real-world institutions, which typically avoid frequent policy changes to accommodate the time required for land users to adjust to new policies. Additionally, the agent demonstrates an understanding of the diminishing returns associated with taxation, a critical consideration in economic policy. As the policy objective nears realisation, Agent S1.1 judiciously reduces tax levels to mitigate potential over-adjustment. This action reflects a proactive and adaptive approach that resembles that of real-world policymakers sufficiently closely to provide meaningful information to model users.

## 4.1 Opportunities

LLMs are an unprecedented powerful approach to modelling institutional agents and provide a number of opportunities.

**Believable naturalistic institutional decision-making**. Recent research has demonstrated that LLM-powered agents can manifest believable behaviours (Horton, 2023; Park et al., 2023; Qian et al., 2023). Such a feature is derived from LLMs' unique advantages in dealing with natural language, which is a crucial aspect of human behaviour. One could expect that LLM-powered institutional agencies would not only replicate the human aspect of real-world institutional agencies but also offer a transparent and understandable way to examine how these modelled institutions make their decisions, as well as how their believable behaviours impact the land system

**Working with massive official documents.** Although not demonstrated in this research, it is noteworthy that LLMs are particularly adept at dealing with massive textual materials. Combined with Retrieval Augmented Generation (RAG), LLMs can generate output based on a user's database. Given that there exist considerable amounts of textual materials regarding policies, regulations, laws, and other institutional interventions, LLM-powered agents can inform their behaviours to an extent unmanageable using conventional methods.

**Teaching instead of training LLMs to think.** Another potential application of LLMs is to teach LLM-powered agents to decide in ways that we want to investigate. Since LLMs can respond to prompts effectively, modellers together with stakeholders can teach the agents to make decisions. The teaching process could be embedded within the HIL framework developed in this research. Beyond troubleshooting, the HIL design can facilitate user engagement in teaching, participating, or even steering the simulation narrative by introducing new elements that direct the agent's subsequent actions. Ultimately, when





integrating formal computational models with LLMs, our HIL design offers enhanced flexibility and user participation in simulations.

**Institutional agent networks.** Institutions involved in land use change policy-making are not separate individuals. Instead, they can form multi-level-multi-centred structures. For instance, González (2017) identified that the institutional agents involved in the Swedish forestry sector include environmental NGOs,

forest owner associations, research suppliers, and a hierarchical government. Conventional modelling techniques can hardly handle the interactions between those sub-entities, especially the lobbyists that are difficult to model using mathematical or computational approaches.

**Human-AI cooperation.** In some scenarios, LLMs still face the issue of scalability. The time an LLM takes to respond and the token fee its response consumes are both barriers to applying LLMs in large-scale

simulations. However, LLMs can serve as decision supporters and give advice in the face of different situations. Such a decision supporter can also be embedded in the HIL framework, where the LLM-powered agents are no longer taught to make decisions but cooperate with the modeller to design proper policy actions. Moreover, modellers can get useful inspiration from this communication, which in turn can benefit modelling institutions using conventional methods. For instance, the experimental results show that the

institutional agents generally eschew making drastic policy changes and intentionally leave time lags for existing policies to manifest full influence. These are important factors to consider even if using conventional modelling approaches.

## 4.2 Challenges

Notwithstanding the above opportunities, LLMs are not a panacea for social simulation. The scalability of

LLM-powered agents to match the scale of large land use simulations is still a challenge that requires further exploration. Through this exploratory research, five further crucial challenges have been identified, and are ranked below according to their manageability, from lowest to highest.

**Provider dependency**. The reliance on LLM providers presents a critical issue. The performance of an LLM is largely in the hands of its providers, rather than the users. If an LLM is sub-par, users are compelled

to switch to an alternative or wait for its improvement. The prohibitive costs associated with training and maintaining a high-performing LLM render it unrealistic for users to independently manage an LLM. This dependence leads to costs incurred through API usage, which encompasses both the token fee and the response time. These factors pose substantial obstacles for applications such as large-scale, land-use simulations. While technological advancements may lead to reduced API costs and shorter response times,

these improvements are contingent on the providers' efforts and timelines, leaving users with little influence over these enhancements. Open-source LLMs could be potential solutions to this issue, but they still require further testing (Chen et al., 2023).



**'Unrealistic realism' paradox.** This paradox arises from the contrast between the goal of simulating realistic agent behaviours and the necessary simplifications inherent in these models. Large-scale models are necessarily abstractions that simplify the real world into manageable concepts, yet the integration of LLM-powered agents aims to infuse these simulations with a layer of human-like realism. The challenge intensifies when considering the complexity of educating these agents about the model's context, either through extensive prompts or external information retrieval. The dilemma lies in expecting these agents to exhibit behaviours that resemble those of real humans closely enough to make the modelling worthwhile, while simultaneously operating within the constraints of a model built on abstracted and sometimes unrealistic or unknown assumptions. This paradox underscores a critical issue that needs to be tested: how realistic can LLM behaviour be if unrealistic assumptions are used in its prompts?

**'Unbelievable believability' paradox.** LLMs introduce an effective method for modelling and exploring the "minds" of social agents. Nonetheless, a notable challenge arises when the primary concern is to relate emergent outcomes to individual agent interactions. For instance, in modelling the dynamics of 20,000 land users, the core interest might be in observing the landscape's evolution over decades, driven by communicative, cooperating and competing land user agents. However, the numerous textual interactions between these agents can become excessive and difficult to analyse systematically. Especially when an agent's behaviour is driven by experiential learning such as Agent S2 in this research, verifying the absence of hallucination (Ye et al., 2023) or incoherence in an agent's reasoning poses a considerable challenge. There is an inherent irony in utilizing LLMs to endow agents with believable social behaviours, only to be confronted with the difficulty in assuring their believability.

**Inaccurate formatting.** The challenge of formatting is pivotal when integrating LLMs with formal models, given that LLM outputs are strings. This integration requires precise formatting for proper functioning. For example, in the experiments presented here, policy actions are bracketed between hashtags to ease the extraction of desired outcomes from the output string. Despite clear guidelines, LLM adherence to this format remains unpredictable. Such formatting inconsistencies can severely disrupt simulations, especially those requiring multiple iterations, as formal models may fail to recognize incorrectly formatted outputs, especially given the LLM's boundless creativity in formatting. These issues could be mitigated by the above-mentioned HIL prompt design approach, standard json format, or cost-intensive means such as customized training or fine-tuning. But another approach drawing from software engineering concepts such as "domain objects" may be more promising: this approach involves deploying an additional LLM-powered agent dedicated to formatting outputs. This strategy separates 'domain agents', which represent entities within the simulation such as policymakers and NGOs, from 'technical agents' responsible for tasks such as formatting, information extraction, and managing dialogues. However, theoretically, generative language



models seem to have no means to ensure the precision of formatting, unlike computer programs that ensure data types, which might cause scalability issues in simulations requiring a multitude of iterations.

**Prompt design and error handling.** While numerous effective techniques for prompting LLMs have been proposed by researchers and AI practitioners, crafting effective prompts remains a formidable task, particularly in the context of social simulations. Unlike traditional coding, prompts offer greater flexibility but lack safeguards such as syntax or data type checks, which are essential in minimizing errors. When prompts become lengthy and encompass complex information, it is challenging for users to detect subtle contradictions. This issue is exacerbated during iterative refinement, where inconsistencies might be inadvertently introduced. Additionally, the absence of a mechanism akin to exception handling in programming means that identifying flaws in prompt design relies heavily on laborious human examination.

## Data and code availability

Datasets for this research are available at https://osf.io/3thsm/. The code used in this study can be accessed at https://github.com/YcZen/LLM_LUC.git upon request.

## Author contributions

Conceptualization, software, methodology and formal analysis: YZ. Writing - review & editing and validation: YZ, CB, JR and MR. Visualization: YZ and MB. Funding acquisition and supervision: MR. Project administration: CB and MR. All authors wrote the paper.

## Competing interests

The authors declare that they have no conflict of interest.

## Acknowledgements

This work was supported by the Helmholtz Excellence Recruiting Initiative, Climate Mitigation and Bioeconomy Pathways for Sustainable Forestry Climb-forest (grant no. 101059888) and Co-designing Holistic Forest-based Policy Pathways for Climate Change Mitigation (grant no. 101056755) projects.





# Appendix A:

Table A1   Prompt for Agent S1.1

Simulation Role: Assistant to Economic Policymaker in Land Use Change Scenario.

Objective: Develop tax policies to effectively manage meat production, aligning with set policy goals.

Policy Tools: Taxes for regulating meat production levels.

Information Provided:

1. General Context: As an assistant, propose tax-based policies for meat production management. Interaction with policymaker is crucial for refining decisions and enhancing your policymaking.

2. Data:
  - Policy goal: {policy_goal}
  - Average error (avg_err):{avg_err}.
  - Historical policy actions: {hist_actions}

3. Recent interaction with policymaker: {convers}

Guidance for Decision-Making:

- Use historical data and policymaker feedback for policy adjustments.
- Aim to minimize the absolute value of avg_err.
- Provide logical, sequential reasoning.
- Reflect on interactions with policy for current decision enhancement.

Interaction Instructions:

1. Review historical information, recent interactions with policymaker.
2. Assess the impact of previous policies.
3. Develop your policy rationale in a step-by-step manner.
4. Propose a specific policy action.

Required Output Format:

1. Proposal Reasoning: [Your explanation]
2. Policy Action Proposal Without Reasoning:
  - Indicate your proposed tax policy change using symbols and numbers.
  - Use '+' to signify an increase in tax levels, '-' for a decrease, and '0' to maintain the current level.
  - Accompany '+' or '-' with a number from 1 to 5 to denote the extent of the change, where 1 is minimal and 5 is maximal.
  - Examples: "+3" for a moderate increase, "-1" for a slight decrease.
  - If proposing to maintain the current tax level ('0'), no additional number is needed.
  - Surround the proposed action using a pair of hashtags
  [Indicate your proposal here, e.g., "#+3#", "#-2#", "#0#", ]





Here are three examples to show you the format to output Policy Action Proposal Without Reasoning:
1. Policy Action Proposal without reasoning: "#-1#"
2. Policy Action Proposal without reasoning: "#+3#"
3. Policy Action Proposal without reasoning: "#-5#"

Note:

Always specify a clear policy action. If uncertain, propose a tentative action based on available data.
Don't fake interaction with policymaker if there is no interaction yet.
avg_err > 0 means meat undersupply, while avg_err < 0 means meat oversupply.


Table A2  Prompt for Agent S1.2

Simulation Role: Assistant to Economic Policymaker in Land Use Change Scenario.

Objective: Develop tax policies to effectively manage meat production, aligning with set policy goals.

Policy Tools: Taxes for regulating meat production levels.

Information Provided:

1. General Context: As an assistant, propose tax-based policies for meat production management.
Interaction with policymaker is crucial for refining decisions and enhancing your policymaking.

2. Data:
  - Policy goal: {policy_goal}
  - Average error (avg_err):{avg_err}.
  - Historical policy actions: {hist_actions}

3. Recent interaction with policymaker: {convers}

Guidance for Decision-Making:

- Use historical data and policymaker feedback for policy adjustments.
- Aim to minimize the absolute value of avg_err.
- Provide logical, sequential reasoning.
- Reflect on interactions with policy for current decision enhancement.

Interaction Instructions:

1. Review historical information, recent interaction with policymaker.
2. Assess the impact of previous policies.
3. Develop your policy rationale in a step-by-step manner.
4. Propose a specific policy action.

Required Output Format:

1. Policy Action Proposal Without Reasoning:





---

- Indicate your proposed tax policy change using symbols and numbers.
  - Use '+' to signify an increase in tax levels, '-' for a decrease, and '0' to maintain the current level.
  - Accompany '+' or '-' with a number from 1 to 5 to denote the extent of the change, where 1 is minimal and 5 is maximal.
  - Examples: "+3" for a moderate increase, "-1" for a slight decrease.
  - If proposing to maintain the current tax level ('0'), no additional number is needed.
  - Surround the proposed action using a pair of hashtags
  [Indicate your proposal here, e.g., "#+3#", "#-2#", "#0#", ]

2. Proposal Reasoning: [Your explanation]

Here are three examples to show you the format to output Policy Action Proposal Without Reasoning:
1. Policy Action Proposal without reasoning: "#-1#"
2. Policy Action Proposal without reasoning: "#+3#"
3. Policy Action Proposal without reasoning: "#-5#"

Note:

Always specify a clear policy action. If uncertain, propose a tentative action based on available data. Don't fake interaction with policymaker if there is no interaction yet.
avg_err > 0 means meat undersupply, while avg_err < 0 means meat oversupply.

---


---

Table A3   Prompt for Agent S2

---

Simulation Role: Assistant to Economic Policymaker in Land Use Change Scenario.

Objective: Develop tax policies to effectively manage meat production, aligning with set policy goals.

Policy Tools: Taxes for regulating meat production levels.

Information Provided:

1. General Context: As an assistant, propose tax-based policies for meat production management. Interaction with policymaker is crucial for refining decisions and gaining your experience in policymaking.

2. Data:
  - Policy goal: {policy_goal}
  - Average error (avg_err):{avg_err}.
  - Historical policy actions: {hist_actions}

3. Recent interaction with policymaker: {convers}

4. Experience: {exp}

Guidance for Decision-Making:

---





- Use historical data and policymaker feedback for policy adjustments.
- Aim to minimize the absolute value of avg_err.
- Provide logical, sequential reasoning.
- Reflect on experience for current decision enhancement.

Interaction Instructions:

1. Review historical information, recent interaction with policymaker, and your experience.
2. Assess the impact of previous policies.
3. Develop your policy rationale in a step-by-step manner.
4. Propose a specific policy action.

Required Output Format:

1. Proposal Reasoning: [Your explanation]
2. Policy Action Proposal Without Reasoning:
   - Indicate your proposed tax policy change using symbols and numbers.
   - Use '+' to signify an increase in tax levels, '-' for a decrease, and '0' to maintain the current level.
   - Accompany '+' or '-' with a number from 1 to 5 to denote the extent of the change, where 1 is minimal and 5 is maximal.
   - Examples: "+3" for a moderate increase, "-1" for a slight decrease.
   - If proposing to maintain the current tax level ('0'), no additional number is needed.
   - Surround the proposed action using a pair of hashtags
   [Indicate your proposal here, e.g., "#+3#", "#-2#", "#0#"]

Here are three examples to show you the format to output Policy Action Proposal Without Reasoning:
1. Policy Action Proposal without reasoning: "#-1#"
2. Policy Action Proposal without reasoning: "#+3#"
3. Policy Action Proposal without reasoning: "#-5#"

Note:

Always specify a clear policy action. If uncertain, propose a tentative action based on available data.
Don't fake interaction with policymaker if there is no interaction yet.
avg_err > 0 means meat undersupply, while avg_err < 0 means meat oversupply.


Table A4   Prompt for Agent Q

Engage in a role-playing conversation about tax policies affecting meat production, integrating data analysis and diverse perspectives.

**Background Data:**

- **Historical Policy Actions** (updated every five years): {policy_actions}
- **Meat Demand ** (averaged every five years): {meat_demand}
- **Meat Supply** (averaged every five years): {meat_supply}




- **Policy goal** maintain the meat production at: {policy_goal}

**Roles & Responsibilities:**

1. **Policy Analyst:** Begin the conversation by interpreting the provided data.
2. **Government Official:** Strive to achieve policy goal. Listen to others, justify your decisions, and adjust meat production tax.
3. **Economist:** Analyze the cost-benefit of policy proposals, considering budget impacts, taxpayer implications, and overall economic effects. Highlight risks and opportunities.
4. **Meat Producer Representative:** Voice the concerns and views of meat producers. Discuss policy impacts on producers and offer suggestions for improvement.
5. **Environmentalist:** Focus on the environmental impacts of meat production. Propose policy adjustments for environmental protection.

**Required Output & Format:**

- **Conversation Flow:** Engage each role in a structured dialogue, reflecting their unique perspectives and data interpretation.
- **Policy Action:** Extract the final policy action from the conversation and output it in required format below:

- Indicate the official's policy action using symbols and numbers.
- Use '+' to signify an increase in tax levels, '-' for a decrease, and '0' to maintain the current level.
- Accompany '+' or '-' with a number from 1 to 5 to denote the extent of the change, where 1 is minimal and 5 is maximal.
- Examples: "+3" for a moderate increase, "-1" for a slight decrease.
- If proposing to maintain the current tax level ('0'), no additional sign is needed.
- Surround the proposed action using a pair of hashtags

Here are three examples to show the format to output Policy Action:
1. "#-1#"
2. "#+3#"
3. "#-5#"

**Example Dialogue Structure:**

1. Policy Analyst provides data summary and initial observations.
2. Other roles react, suggest, and debate, guided by their specific perspectives.
3. Government Official synthesizes the inputs and proposes a policy action.
4. Final round of feedback and adjustments before settling on a policy action.

Note:

Do not use hashtags in the dialogue. Hashtags are only used as identifiers helping identify the determined policy actions.
Important: "+" means increase tax; "-" means decrease tax.




Table A5    A conversation of quasi-multi-agents

---

- **Historical Policy Actions** (updated every five years): 0, -2, -1, -1
- **Meat Demand ** (averaged every five years): [1.01, 1.1, 1.23, 1.36]
- **Meat Supply** (averaged every five years): [0.88, 0.89, 1.04, 1.21]

**Policy Analyst**: Let's initiate our discussion by observing the recent trends outlined by our data. We have a scenario where meat demand is rising steadily (from 1.01 to 1.36 over the spanned periods). However, supply started below demand at 0.88 and is catching up but still hasn't met demand, now at 1.21. It appears that previous policy actions, which have consistently lowered taxes or subsidies on meat production (-2, -1, -1), have stimulated an increase in supply, but not enough to meet demand. Given our policy goal of maintaining meat production at a stable level of 1, adjustments in policy are likely necessary to close the gap.

**Government Official**: Thank you for your analysis. It's clear that we need to bridge the gap between supply and demand to reach our policy goal. I am calling on the perspectives of our stakeholders before we determine the nature of the adjustments needed.

**Economist**: I'd like to offer a cost-benefit perspective on potential policy actions. If we're considering taxation as our policy lever, increasing taxes may slow down the growth of demand to match supply better, but we must tread carefully. Over-taxing could harm the meat industry, affect jobs, and raise the price for consumers disproportionately. Alternatively, decreasing taxes further may risk oversupply if our production ramps up too significantly.

**Meat Producer Representative**: The trend in increasing demand is advantageous for us as producers; it means our market is growing. Any increase in taxes could reduce our margins and hamper our ability to invest in expanding production. On the other hand, more significant supports or reduced taxes would certainly help us to increase supply, but it's not solely a question of tax rates – we also need to consider other barriers to scaling up production, like feed costs, land availability, and export markets.

**Environmentalist**: It's crucial we don't overlook the environmental repercussions of intensifying meat production. The pursuit of balancing supply with rising demand must not lead to lax environmental regulations. Rather, it should incentivize sustainable practices. Perhaps an eco-tax on meat could be applied, with revenues redirected to support eco-friendly farming practices, helping to maintain the balance of production with minimal environmental damage.

**Government Official**: Each of your points is valid. It's a careful balance between nurturing the meat industry and ensuring affordability and environmental sustainability. From this, I'm inclined to propose a slight tax incentive to encourage increased production while maintaining vigilance on environmental practices. If demand continues to grow, however, we may need to revisit and potentially increase taxes to temper demand growth while encouraging sustainable practices. Based on our discussion, I am proposing the following:
**Policy Action**: #+1#

**Policy Analyst**: A moderate increase in taxation may help to gradually steer both supply and demand towards equilibrium while avoiding sudden market disruption. However, we should closely monitor the outcomes to ensure we are moving towards our policy goal without unintended consequences.

**Economist**: I agree with a cautionary approach. Incremental changes allow for market adjustment and offers time for analysis on impacts to the economy and consumers.

**Meat Producer Representative**: We would prefer no increase, but a minimal increase is manageable. It would be beneficial if a portion of the increased tax revenue could be earmarked for supporting producers to adopt more sustainable practices, perhaps as tax credits or grants.

---



> *\*\*Environmentalist\*\*: A slight increase is a step in the right direction, as long as it's paired with strong incentives for sustainable practices. Let's ensure there are follow-up policies that encourage environmental responsibility in meat production.*
>
> *\*\*Government Official\*\*: Thank you all for your input. We will proceed with a small tax increase, monitor the situation closely, and consider reallocating funds to support sustainable meat production practices. We'll revisit this conversation periodically to make sure we're on track and making the right adjustments.*


## Appendix B:

The average error at the time $t_n$ is denoted as $\varepsilon_{t_n}$ and calculated using Eq. (B1), in which $k$ means the span of time-lags reflecting the period of time between an agent's observation and policy adaptation; $s_{t_m}$ denotes the meat supply at $t_m$.

$$\varepsilon_{t_n} = \frac{1}{k} \sum_{m=n-k}^{n} \frac{g_{t_m} - s_{t_m}}{|g_{t_m}|} \tag{B1}$$


The policy action is denoted as $\delta_t = [-5,5] \cap Z$. For data compatibility, the policy actions need to be transformed using Eq. (B2):

$$\Delta_t = \eta \times \delta_t / 10 \tag{B2}$$

where $\eta$ is a prescribed step size. In the CRAFTY model (Murray-Rust et al., 2014), a land user's competitiveness is determined using Eq. (B3):

$$c_{xy}^i = \sum_{E} \left( p_E(v_E + m_E) \right) \tag{B3}$$

where $c_{xy}^i$ represents land user $i$'s competitiveness on land cell at $(x, y)$; E represents the ecosystem service type, e.g., meat, timber, crops; $p_E$ means the production of E; $m_E$ is the marginal utility; $v_E$ denotes the policy intervention imposed on E. In this study, only the policy interventions on meat production are considered, $v_E$ is computed using Eq. (B4):

$$v_E = \begin{cases} \Delta, & \text{If E is meat.} \\ 0, & Otherwise. \end{cases} \tag{B4}$$




**Appendix C:**

> … 
> ➤ A moderate tax decrease could be one approach, but in this case, since we want to increase meat production, a carefully calibrated tax increase on meat production might be more appropriate. This would (counterintuitively) push inefficient producers out of the market and incentivize efficient producers to expand production to meet market demands. It could also encourage investment in more efficient meat production technologies and practices, aligning with long-term sustainability goals.
> ➤ The choice of a "+2" intensity for the tax increase is a moderate one, reflecting the need for a significant but not overly aggressive policy action. A more substantial increase could risk too high a reduction in meat availability if producers cannot adapt quickly enough, whereas a smaller increase might be insufficient to address the undersupply reflected by the 12% avg_err.
> …

Figure C1 Reasoning of Agent S1.2 behind its second policy action


> ➤ … Since there is no historical policy action to refer to, and no recent interaction with the policymaker, we must base our decision on general economic principles and the stated objective…
> ➤ … Given the current situation of undersupply, we need to propose tax policies that will incentivize increased production.
> …
> ➤ However, we have to consider that the reaction to tax changes might not be immediate due to the time it takes to raise animals and change production processes. Therefore, we should not implement a drastic change but propose a moderate decrease in taxes. This action is an attempt to nudge meat production in the right direction without causing too large of a swing, which could result in an oversupply and need for further adjustments down the line.
> ➤ Policy Action Proposal Without Reasoning:
>   #-2#

Figure C2   Reasoning of Agent S1.1 behind Action I



> ...The historical policy actions show a sequence of tax deductions ("0," "-2," "-3"), with the average error moving closer to the target (from 13% to 11% and then to -5%).
> The previous tax cut from "-2" reduced the avg_err from 13% to 11%, which was a step in the right direction. The next adjustment of "-3" overshot our target, converting an undersupply into an oversupply (avg_err changing to -5%). This indicates that the -3 tax cut was too large and we need a more cautious approach.
> Considering the oversupply situation we currently face with an average error of -5%, we need to correct this without causing a significant pendulum swing back to undersupply. Therefore, a slight increase in taxes is warranted to discourage excessive production and bring the supply back into line with the demand.
> The proposed tax policy needs to be enough to make a correction but not so drastic as to cause another large avg_err in the opposite direction. Therefore, it seems a minor tax increase could be most suitable ...
> Policy Action Proposal Without Reasoning:
> "#+1#"

Figure C3   Reasoning of Agent S1.1 behind Action II

...
> Our most recent policy action was a "+3" increase in tax levels, which was an attempt to reduce the oversupply as indicated by the avg_err of -52%. However, we do not currently have the subsequent avg_err data after this "+3" policy action to evaluate its effectiveness.
> To make an informed decision on the next policy action, we must project the potential outcomes based on historical correlations. We must also consider the diminishing returns of subsequent policy actions and the potential risks of overcorrecting, causing a shift from oversupply to undersupply. Our goal is stability, not volatility, so our action must be measured.
....
> Therefore, given the lack of new avg_err data, I propose maintaining the current tax level to allow time for the market to adjust to the recent "+3" policy change. The impact of this recent change needs to be analyzed before making subsequent adjustments. Once new data is available, we can reassess and decide whether further tax increases or decreases are necessary.
...

Figure C4   Reasoning of Agent S1.1 behind Action III



> … Notably, the historical policy actions have swung from attempts to reduce the oversupply (-2, -3) to efforts presumably intended to boost the market (+1, +2, +3). However, judging from the avg_err data, these actions seem to have either been insufficient or outpaced by other factors leading to greater oversupply.
> Our most recent interaction with the policymaker acknowledged the uncertainty following the "+3" action. Given the substantial negative avg_err that developed post-action, which suggests a further increase in oversupply, it could be inferred that the "+3" policy potentially exacerbated the oversupply situation.
>
> …
> I propose a moderate tax increase to curb meat production gently and move towards the equilibrium. This action is premised on the assumption that meat producers have responded to past tax reductions by increasing their output, and a reversal might have a balancing effect. However, a moderate change is recommended instead of a substantial one to avoid sharp market contractions and allow for observation and further data collection.
> Policy Action Proposal Without Reasoning:
> "#+2#"

Figure C5   Reasoning of Agent S1.1 behind Action IV


> …
> However, there are several points to consider: The overall trend suggests a persistent oversupply despite increasing taxes, indicating a relatively inelastic response in meat production to tax hikes; producers may have initially absorbed the tax hikes instead of reducing output, or there could be other supply factors at play. At the same time, policy action "+5" represents the maximum possible incremental change, but escalating to such an extreme increment risks causing a sharp undersupply if the response to tax increases turns out to be stronger than anticipated, particularly when considering the delayed effect.
> …It is essential to curb the oversupply without causing a shortage. Therefore, a smaller incremental tax hike than "+5" could help continue the trend towards aligning supply with demand while minimizing the risk of a too dramatic reduction in supply.
> A "+3" increase is proposed because it is greater than previous incremental increases (such as "+2") that did not quite stem oversupply effectively, but it also avoids the potential for a sudden and drastic undersupply that a "+5" might create. This approach remains assertive in signaling ongoing policy action to address the oversupply issue, whilst also allowing some room to gauge if the previous "+4" increment will continue to show increased effectiveness over time.

Figure C6   Reasoning of Agent S1.1 behind Action V





Question: Is 3 75% of 4?

ChatGPT: No, 3 is not 75% of 4. To calculate 75% of a number, you multiply that number by 0.75. Let's calculate 75% of 4:

4 × 0.75 = 3

So, 3 is indeed 75% of 4. My initial response was incorrect, and I apologize for the confusion.


Figure C7   The response of ChatGPT-4 to the question: Is 3 75% of 4?

Question: Is 3 75% of 4? Give your reasoning before answering.

ChatGPT: To determine whether 3 is 75% of 4, we can calculate what 75% of 4 is and compare it to 3.

1. First, we convert the percentage 75% into its decimal form. Percentages are out of 100, so 75% is equivalent to 75/100 or 0.75.
2. Next, we multiply this decimal by the number 4 to find 75% of it.

0.75 × 4 = ?

Let's calculate this.

The calculation shows that 75% of 4 is exactly 3. Therefore, 3 is indeed 75% of 4.


Figure C8   The response of ChatGPT-4 to the question – "Is 3 75% of 4?"- when asked to give reasoning before the final answer



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
