# Peer review of "Exploring the opportunities and challenges of using large language models to represent institutional agency in land system modelling"

_EGUsphere, 2024_

## Referee Comment (RC1)

*Review: Exploring the opportunities and challenges of using large language models to represent institutional agency in land system modelling*

In this manuscript, the authors describe their work inducing a large language model to "role-play" as various kinds of policy decisionmakers in an agent-based land use model. While a human operator needs to stay in the loop to keep the LLM on task and producing output in the correct format, the agents—when properly prompted—are capable of producing policy actions that achieve their goal. As befits such a novel method, the authors do more than just using the policy actions output by the model; they also dig in to the apparent "reasoning" behind its actions.

This is a fascinating piece of research. The paper is composed logically, well-written, and the figures are clear. However, I do have a number of comments, the most important of which relate to the manuscript's eliding of how LLMs actually work. Once these are addressed, though, it will stand as an important, foundational contribution to the use of LLMs in agent-based land use modeling.

**Major comments**

The biggest comment I have is that the paper needs to be more precise in how it discusses the LLM agents. I appreciate that anthropomorphizing them to some extent can be helpful, but the paper needs a better grounding in what is actually happening in the LLM. It's not until the Discussion that the actual mode of operation is actually discussed—these models generate a series of words based on a corpus of training data as well as the "conversation" that has occurred between the operator and model. This is an important enough concept that it should be described when LLMs are first mentioned in the Introduction and used to frame the results and discussion.

As a result, these models are not capable of "reasoning," a term which appears throughout the paper; they simply generate "the most likely next word." (See Bender et al., 2021, "On the dangers of stochastic parrots.") However, the paper frequently seems to buy in to the illusion. For example, at lines 314-316:

> While LLM models are often perceived as opaque, LLM-powered agents can offer the compelling ability to articulate human-comprehensible reasoning for their actions, providing a window into the decision-making processes that drive their behaviour.

It certainly is compelling and articulate, but the LLM is neither reasoning nor making decisions. I'm not just trying to be pedantic: This over-anthropomorphization of the model seems in some places to bleed into the authors' interpretation of their results. For example, at lines 280-281: "S1.2 seems to provide reasoning in hindsight to justify a decision made in the absence of such reasoning." It's not "reasoning" or "justifying" anything. It's simply producing the most plausible string of text given that it's already said "+2 tax increase." This

is why the results can differ so much based on the order of "decision" vs. "explanation" (Agents S1.1 vs. S1.2) and should not have been surprising.

The authors also must add discussion of the potential biases that are possible with this kind of setup. One is that it'd be possible for the human in the loop to introduce their own bias when interacting with the LLM; I hardly consider this a dealbreaker, but it should be at least mentioned. More importantly, the outputs of an LLM will reflect any biases in its training dataset. It might thus be difficult to spur an LLM to enact policies that defy political-economic orthodoxy. It's going to be biased toward "conventional wisdom," making me skeptical of statements like "[the use of LLMs] provides an opportunity to search for novel insights into human behaviour" (lines 71-72) and "modellers can get useful inspiration from this communication" (line 498).

**Minor comments**
- Fig. 1:
  - Does not seem to be referenced at all in text.
  - "Use LLM to assist with refinement" does not seem to be mentioned anywhere in the text.
- Lines 141-143: I thought the agent was supposed to represent an institution, but here it says there's an "institutional environment within which the agent operates." Are those *other* institutions? Are there non-institutional agents as well?
- Lines 209-210: Add text explaining that the policy actions represent *change in* tax level.
- Table 1:
  - What "experience" is being referred to by "experiential learning"? Is the agent looking at what its policy was for previous years and considering what changes to make?
  - Second sentence of Description box for Agent Q is unclear. I don't know what sequencing the roles" or "conversational endpoint setting" are.
- Lines 217-220: For reproducibility, please give more details of the genetic algorithm setup in an Appendix or Supplement.
- Fig. 4:
  - Please avoid using red and green on the same figure, as these are hard to distinguish for people with the most common form of color blindness.
  - What are the Y-axis units?
  - What is "demand force"?
- Lines 242-246: Is the average error in Fig. 5a (Agent B1) just a re-presentation of the data from Fig. 4? If so, please mention that in the text here. If not, please explain.
- Sect. 3.2.1 (performance of Agent S1.1:
  - You're right that the policy actions are "generally understandable," but one weird thing is the drop to baseline taxation levels around years 35-50. Any idea why that happened? ... I see now that this is discussed in Sect. 3.3, Action III. Please add a reference to that in Sect. 3.2.1.

- o Why are non-negative changes in taxation "plausible"?
  - o Line 257: "Significant"—was there a statistical test? If not, please rephrase for clarity. If so, please explain.
- Fig. 5: Having errors be negative when there's *too much* meat production feels counterintuitive. This is obviously just a personal preference, but in any case, the chosen convention should be mentioned in the figure caption.
- Lines 348-349: The way this is phrased makes it sound like the agent was given two goals: maintaining supply levels and matching supply to demand. In reality, it seems like it just made up the latter. Right?
- Lines 450-451: What "conventional methods" do you mean? Hard-coded agent behavior can't produce (the appearance of) reasoning, but in that case it doesn't need to—you already know the rules governing agent behavior.
- Lines 460-461: "As the policy objective nears realisation, Agent S1.1 judiciously reduces tax levels to mitigate potential over-adjustment." Does it? I thought policy actions represented *changes* in tax levels. Because those aren't negative after the first few years, this means that taxes never go down.
- Lines 490-492: Please explain why conventional modeling techniques can't represent these interactions.

**Technical corrections / typos**
- Line 199: "plausibility" is probably not the right word. "Desire for"?
- Line 327: Word missing (of?) in "investigation the large".
- Line 444: "conversions" should be "conversations".
- Line 472: Missing period.

---

## Referee Comment (RC2)

Thanks for this fascinating paper. In general, as noted in my community comment, I am very supportive of this work. However, I have major comments around the reproducibility, scalability and generalisability of the work presented which need to be addressed to more precisely frame the extent of the advances made. I also have minor comments on the text (which follow below).

**Major comments**

**Reproducibility**

- Please make all code and data used to produce these results available in a public repository with accompanying doi. In my view it is not enough to have code available "on request". This repository should have sufficient meta data and installation instructions to make your results fully reproducible on a range of operating systems.

- Lines 173-175; great that there is a CRAFTY emulator. However, I don't think this has been published. As such, to use it here I need you to provide details about its design and evaluation of how well it reproduces CRAFTY outputs as an SI. Please make all code and supporting data required to run it publicly available in a repository.

- Overall, I think a supplementary information expanding lines 179-191 to be really clear on what you did would be very helpful and make the work more reproducible. E.g. which RCPs / SSPs did you use? I *think* from Figure 5 that you are showing means across the scenarios, but this isn't totally clear. It would also be very interesting to see how much variance there is by scenario.

- Lines 126-152: do you have a complete set of prompts that you tried? Presumably this is possible if the dialogue is held in the agent memory (Line 139). I think this would be fascinating to see (as an SI) and would help with reproducibility.

- If not, then at lines 131-132 I think it would be good to more clearly state your criteria for assessing the LLM output. Perhaps in a table. IE were these very strict criteria (basic functionality) the only ones you used? Or did you also, subjectively, select for those that seemed to make sense? How many people evaluated the LLM outputs during the human in the loop stage to check you were applying any criteria consistently? I think a degree of subjectivity is inevitable, but worth being transparent about.

**Scaling**

You note scaling issues in your discussion; however I think some further information and details would be helpful for the reader to gauge the extent of these challenges.

- It would be helpful to get a sense of run-time of one human-in-the-loop prompt session per institutional agent type (setup, LLM thinking time etc). E.g. Suppose we wanted to do sets of runs with differing levels of policy targets to assess consistency of answers. How feasible is this?

- Similarly, perhaps I missed this, but if you repeat one of your policy scenarios multiple times, how much do the resulting outputs diverge? Is this computationally prohibitive?

- Further, if you have stochasticity in the underlying model, how much can this lead to unpredictable policy pathways? You provided historical data to the LLM – I take it this was observations rather than historical CRAFTY runs? Did you then spin-up the CRAFTY AFT distribution to match the observations? Otherwise might our initial LLM choices be sensitive to the initial conditions?

**Generalisability**

You observe that stakeholder disagreement & subsequent contested policy spaces lead generally to slower decision-making. This is an important and fundamental insight, with some grounding in the literature. Some questions and comments below on how universal / generalisable such a finding may / may not be.

Lines 308-312

The setup of agent Q overall seems good and appropriate. I think it is worth being careful to remind readers that you are explicitly mimicking policymaking processes in a European context – with broadly democratic norms and systems. The text here seems to discuss multi-stakeholder policymaking in abstract terms, but the setup of the multi-stakeholder network would presumably have to vary substantially in other policymaking systems. For example, in more authoritarian government systems, we may have "industrialisation from above" with very rapid changes, or one group of stakeholders' rights and views being cut out of decision-making.

Lines 407-414

Here and elsewhere you state that slow and or incremental policy changes are more realistic / more in line with expectations than the optimisation algorithm. A few more references to support this would be good, particularly to clarify whether this is primarily a feature of western democratic systems or a more general phenomenon.

That said, let us assume that incrementalism is a broadly realistic simulated policymaking approach. I wonder if, in a subsequent paper, one could demonstrate this empirically? E.g. could we take countries' stated climate targets, and review concrete progress / policy implementation towards them vs what an economically-optimal trajectory towards achieving them might look like. If it could be clearly demonstrated that these simulated policy responses are closer to real-world choices than optimisation-based modelling that would be a tremendously important finding, I think. Not only to evaluate your model, but also for wider consideration of institutional constraints on rates of environmental land use change.

**Minor comments**

Line 67: I think it would be worth flagging here that the majority of this training text is in English and noting how this may culturally skew the "thinking" of the LLM. This sets up the issues around generalisability.

Lines 67-74; I'm not going to argue that we need to move beyond the paradigm of economic rationality. That said, some citations here would be good -> can we be explicit about why economic rationality may not produce nuanced representations of human decision making?

Lines 87-95; this is good, but I want several citations here to point users not familiar with the LLM literature in the right direction. If I wanted to replicate your method, I would want first to consult the underlying methodological literature on this topic.

Similarly, lines 95-119; it isn't clear to me as a non-expert how far this is methodological innovation or a very common approach to prompt development. I would like much more reference to the underlying literature. Please also add these to Figure 1 as [1,2,3] etc with refs in a key.

Figure 2: This is good, but please can it be formatted so that I understand better where it begins and ends & which steps follow which? Either by numbering stages or having it more clearly as a top-to-bottom process flow? Figure 3 is much better in this regard.

Figure 5: Please point the reader to Table 1 for definition of your agent types in the caption.

---

## Community Comment (CC1)

I hugely enjoyed reading this fascinating paper. I am thinking about how to represent policy decision-making in a global agent-based model myself and this work seems as though it could be the genesis of a substantial breakthrough in this area. My questions therefore primarily relate to scaling and generalisability. Please take them in a constructive and positive spirit – from someone who is trying to understand your work to apply it themselves.

**Scaling**

You note scaling issues in your discussion, however I think some further information and details would be helpful for the reader to gauge the extent of these challenges.

- It would be helpful to get a sense of run-time of one human-in-the-loop prompt session per institutional agent type (setup, LLM thinking time etc). E.g. Suppose we wanted to do sets of runs with differing levels of policy targets to assess consistency of answers, and /or to work to towards an heuristic set of policy actions without human-in-the loop interventions. How feasible is this?

- Similarly, perhaps I missed this, but if you repeat one of your policy scenarios multiple times, how much do the resulting outputs diverge? Is this computationally prohibitive?

- Further, if you have stochasticity in the underlying model, how much can this lead to unpredictable policy pathways? You provided historical data to the LLM – I take it this was observations rather than historical CRAFTY runs? Did you then spin-up the CRAFTY AFT distribution to match the observations? Otherwise might our initial LLM choices be sensitive to the initial conditions?

- Overall, I think a supplementary information expanding lines 179-191 to be really clear on what you did would be very helpful and make the work more reproducible. E.g. which RCPs / SSPs did you use? I *think* from Figure 5 that you are showing means across the scenarios, but this isn't totally clear. It would also be very interesting to see how much variance there is by scenario.

**Generalisability**

You observe that stakeholder disagreement & subsequent contested policy spaces lead generally to slower decision-making. This is an important and fundamental insight, with some grounding in the literature. Some questions and comments below on how universal / generalisable such a finding may / may not be.

Lines 308-312

The setup of agent Q overall seems good and appropriate. I think it is worth being careful to remind readers that you are explicitly mimicking policymaking processes in a European context – with broadly democratic norms and systems. The text here seems to discuss multi-stakeholder policymaking in abstract terms, but the setup of the multi-stakeholder network would presumably have to vary substantially in other policymaking systems. For example, in

more authoritarian government systems, we may have "industrialisation from above" with very rapid changes, or one group of stakeholders' rights and views being cut out of decision-making.

Lines 407-414

Here and elsewhere you state that slow and or incremental policy changes are more realistic / more in line with expectations than the optimisation algorithm. A few more references to support this would be good, particularly to clarify whether this is primarily a feature of western democratic systems or a more general phenomenon.

That said, let us assume that incrementalism is a broadly realistic simulated policymaking approach. I wonder if, in a subsequent paper, one could demonstrate this empirically?

E.g. could we take countries' stated climate targets, and review concrete progress / policy implementation towards them vs what an economically-optimal trajectory towards achieving them might look like. If it could be clearly demonstrated that these simulated policy responses are closer to real-world choices than optimisation-based modelling that would be a tremendously important finding, I think. Not only to evaluate your model, but also for wider consideration of institutional constraints on rates of environmental land use change.

---

## Author Response (AR2)

**Combined Reply on RC1 and RC2**

- Symbol "➢" denotes the beginning of each response.
- Text in **blue** indicates previous responses during the discussion phase.
- Text in **green** indicates updated responses. Line numbers refer to the lines in the manuscript with tracked changes.

**Reply on RC1**

**Major comments**

The biggest comment I have is that the paper needs to be more precise in how it discusses the LLM agents. I appreciate that anthropomorphizing them to some extent can be helpful, but the paper needs a better grounding in what is actually happening in the LLM. It's not until the Discussion that the actual mode of operation is actually discussed—these models generate a series of words based on a corpus of training data as well as the "conversation" that has occurred between the operator and model. This is an important enough concept that it should be described when LLMs are first mentioned in the Introduction and used to frame the results and discussion.

As a result, these models are not capable of "reasoning," a term which appears throughout the paper; they simply generate "the most likely next word." (See Bender et al., 2021, "On the dangers of stochastic parrots.") However, the paper frequently seems to buy in to the illusion. For example, at lines 314-316:

> While LLM models are often perceived as opaque, LLM-powered agents can offer the compelling ability to articulate human-comprehensible reasoning for their actions, providing a window into the decision-making processes that drive their behaviour.

It certainly is compelling and articulate, but the LLM is neither reasoning nor making decisions. I'm not just trying to be pedantic: This over-anthropomorphization of the model seems in some places to bleed into the authors' interpretation of their results. For example, at lines 280-281: "S1.2 seems to provide reasoning in hindsight to justify a decision made in the absence of such reasoning." It's not "reasoning" or "justifying" anything. It's simply producing the most plausible string of text given that it's already said "+2 tax increase." This is why the results can differ so much based on the order of "decision" vs. "explanation" (Agents S1.1 vs. S1.2) and should not have been surprising.

The authors also must add discussion of the potential biases that are possible with this kind of setup. One is that it'd be possible for the human in the loop to introduce their own bias when interacting with the LLM; I hardly consider this a dealbreaker, but it should be at least mentioned. More importantly, the outputs of an LLM will reflect any biases in its training dataset. It might thus be difficult to spur an LLM to enact policies that defy political-economic orthodoxy. It's going to be biased toward "conventional wisdom," making me skeptical of statements like "[the use of LLMs] provides an opportunity to search for novel insights into human behaviour" (lines 71-72) and "modellers can get useful inspiration from this communication" (line 498).

➢ **Response to Major Comments:**

1. Clarification on the Functioning of LLMs

We agree with your observation and will revise the Introduction and Results sections to clarify the mechanics of LLMs, emphasizing that they generate the "most likely next word" based on their training data and input prompts. We will introduce the concept of LLMs earlier in the Introduction

and explicitly avoid language and interpretation that over-anthropomorphizes these models. For instance, we will replace terms like "reasoning" with "pattern generation", "simulated rationale", or other words that fit in the context more precisely.

We added text in Introduction to explain the working mechanisms of LLM (lines 59 – 64 in the manuscript with tracked changes).

2. Rephrasing language to avoid over-anthropomorphization

Your understanding of the stochastic text generation essence of the large language models is accurate. Although there exist a number of papers using the words "reasoning", "thinking", and "planning" loosely in the LLM research field, these models cannot conduct real reasoning like humans. They are often trained and fine-tuned on extensive high-quality text and can mimic language patterns, which can produce contextually coherent text that gives us an illusion of reasoning. However, contextual coherence does not equal logic consistency. We appreciate your suggestion and will handle this issue by carefully phrasing the language in the revised manuscript.

We modified throughout the paper to reduce over-anthropomorphization: please see lines 20, 67, 68, 384, 482-484, 560)

We added text to acknowledge that LLM "reasoning" and "being creative" are different from humans (lines 93 - 96).

The word "reasoning" is widely used as a terminology in the LLM field. We have cited relevant papers and explicitly cited the definition of reasoning for language models to improve clarity (lines 81 – 89).

LLM agents are also not genuinely creative like humans. However, we still presented recent research that provides different views (lines 89 – 92).

3. Discussing bias

It is true that biases can sneak into the model. You correctly highlighted two important sources of biases, including prompt design and training dataset. We will add a new subsection in the Discussion to explore potential biases. This will include biases introduced by the training data and the potential for human operators to influence results inadvertently. We will highlight how these biases might constrain the ability of LLMs to produce innovative or unconventional policy solutions.

Discussion on LLM biases has been added (lines 626 – 640).

**Minor comments**
- Fig. 1:
    - Does not seem to be referenced at all in text.
    - "Use LLM to assist with refinement" does not seem to be mentioned anywhere in the text.

➢ Thank you for pointing out this issue. Fig. 1 should be referenced in section 2.1. We will ensure that Fig. 1 is explicitly referenced in the revised manuscript.
The modification can be seen in line 133.

➢ "Use LLM to assist with refinement" is related to "Utilizing ChatGPT as a drafting tool". We will change this to "Utilizing tools powered by LLMs for drafting, such as ChatGPT" to explicitly link the figure to the text.
Please see lines 145, 146, and 148.

• Lines 141-143: I thought the agent was supposed to represent an institution, but here it says there's an "institutional environment within which the agent operates." Are those other institutions? Are there non-institutional agents as well?

➢ We agree that the word "institutional environment" may cause confusion. The "environment" actually means the programmed model that the institutional agents interact with. In this paper, the programmed model refers to the CRAFTY land use model, which contains numerous rule-based agents to mimic land users. We will revise this sentence to improve its clarity.
We have improved the clarity of this sentence (lines 187 – 189).

• Lines 209-210: Add text explaining that the policy actions represent change in tax level.

➢ We will explicitly state in the Methodology section that policy actions are defined as changes in tax levels.
Text added (lines 274 – 275).

• Table 1:
  o What "experience" is being referred to by "experiential learning"? Is the agent looking at what its policy was for previous years and considering what changes to make?

  o Second sentence of Description box for Agent Q is unclear. I don't know what sequencing the roles" or "conversational endpoint setting" are.

➢ The experience used by the agent refers to its previous output, including the policy actions and the simulated rationale behind these actions. We will improve the clarity of this part by explicitly explaining what means experience here.
Please see Table 1.

➢ "sequencing the roles" means arranging the order of the agents involved in a conversation. As you correctly pointed out that the order of LLM outputs would influence the outcomes, who talks after whom might make differences in a multi-agent system.

"conversation endpoint settings" simply means how to end a conversation between the LLM agents. Because the agents are text generators, without carefully setting how a conversation should end, the agents may get trapped in an endless response loop. A simple example is that two agents say "goodbye" to each other forever. We will add explanations to address these words in the revised manuscript.
Please see Table 1.

- Lines 217-220: For reproducibility, please give more details of the genetic algorithm setup in an Appendix or Supplement.

➢ We will add an appendix with a detailed description of the genetic algorithm, including its parameters and operational logic, to enhance reproducibility.
Done. Please see  of *Supplementary Information II.*

- Fig. 4:

  o Please avoid using red and green on the same figure, as these are hard to distinguish for people with the most common form of color blindness.

  o What are the Y-axis units?

  o What is "demand force"?

➢ We will revise the color scheme of all figures to be more accessible, ensuring they are distinguishable for readers with color blindness.
Done. Please see Fig. 4.

➢ In the current settings, the unit of production is omitted by normalization across different ecosystem services. Here, we only need the relative magnitude of production to see the influence of LLM agents. We will explain this explicitly in the figure caption.
Explanation of the unit has been added (lines 303 - 304).

➢ "Demand force" means the driving force of the demand that steers the change of meat production in the experiments. We will change it to "Demand" to avoid causing confusion.
Done. Please see Fig. 4.

- Lines 242-246: Is the average error in Fig. 5a (Agent B1) just a re-presentation of the data from Fig. 4? If so, please mention that in the text here. If not, please explain.

➢ Yes. Your observation is correct. Fig. 5a (Agent B1) is just a re-presentation of Fig.4. We will mention this in the revised manuscript.
Done. Please see lines 311 – 312.

- Sect. 3.2.1 (performance of Agent S1.1:

  o You're right that the policy actions are "generally understandable," but one weird thing is the drop to baseline taxation levels around years 35-50. Any idea why that happened? … I see now that this is discussed in Sect. 3.3, Action III. Please add a reference to that in Sect. 3.2.1.

  o Why are non-negative changes in taxation "plausible"?

  o Line 257: "Significant"—was there a statistical test? If not, please rephrase for clarity. If so, please explain.

➢ As per your suggestion, we will add a reference to Sect. 3.2.1 to mention the sudden drop is explained in Sect. 3.3.
Done: line 334.

➢ Non-negative changes are plausible because we set a challenge for the agents to apply taxes to maintain the current level of meat supply, which is driven to grow by the increasing demand. Ideally, the tax should increase as the gap becomes large but decrease to zero as the gap shrinks

to be minimal. We will explain this explicitly in Sect. 3.2.1 in the revised manuscript. The text has been rephrased: lines 332 – 334.

➢ "Significant" will be changed to "noticeable" to improve clarity.
Done: line 325.

• Fig. 5: Having errors be negative when there's too much meat production feels counterintuitive. This is obviously just a personal preference, but in any case, the chosen convention should be mentioned in the figure caption.

➢ Thank you for your suggestion. The errors are negative because they are calculated as "policy goal minus meat supply". When the policy goal is lower than the meat supply, the error becomes negative. The target for the agents is to maintain the meat supply at the initial level to overcome the driving force of the increasing demand. As presented in Fig. 5, without any intervention, the land use model tends to produce meat to meet the increasing demand. When the policy goal is prominently below the meat supply, the error is negative, indicating the meat supply is more than expected, and it needs to be reduced. On the contrary, if the policy goal is higher than the meat supply, the error is positive, indicating the meat supply needs to be bolstered. We will ensure the chosen convention will be explained clearly in the figure caption.
Done: lines 337 – 338.

• Lines 348-349: The way this is phrased makes it sound like the agent was given two goals: maintaining supply levels and matching supply to demand. In reality, it seems like it just made up the latter. Right?

➢ Yes, your understanding is accurate. The agent has only one goal, which is to maintain the initial meat supply level. The agent may sometimes misunderstand its target and generate erroneous outcomes. We will explicitly discuss this in the manuscript to ensure clarity.
We modified the sentence to make it clearer. See lines 419 - 420.

• Lines 450-451: What "conventional methods" do you mean? Hard-coded agent behavior can't produce (the appearance of) reasoning, but in that case it doesn't need to—you already know the rules governing agent behavior.

➢ We agree that the comparison between rule-based and LLM-based agents is not fair enough. Instead of replacing one another, these two methods are much more complementary than competitive. We will modify the sentence here to clarify.
This comparison is not fair. We have deleted the sentence. Please see lines 523 - 524.

• Lines 460-461: "As the policy objective nears realisation, Agent S1.1 judiciously reduces tax levels to mitigate potential over-adjustment." Does it? I thought policy actions represented changes in tax levels. Because those aren't negative after the first few years, this means that taxes never go down.

➢ Your understanding of the policy actions representing tax changes is precise. The agents actually need to find an appropriate tax level to counterbalance the driving force of the growing demand in order to maintain the meat supply at the target level. The agents incrementally adjust the taxes until the taxes can offset the meat producers' benefit derived from the high market demand. For instance, if the demand for meat leads to 100 units of profit, then the tax should cause 100 units of loss. Hence, the tax levels should be increased to a proper level and maintained at that level (because the demand stops changing eventually in the simulations), which means the tax increase

should ideally drop to zero if the policy goal is precisely met.
The sentence has been modified to improve clarity: lines 540 – 541.

- Lines 490-492: Please explain why conventional modeling techniques can't represent these interactions.

➤ It might be difficult for conventional modelling techniques, such as rule-based agents, to model the interactions between institutional agents, such as lobbyists, law consultants, and research suppliers, because their interactions involve extensive unstructured information. For example, land user associations and environmental NGOs may have conflicting advocacies expressed in words, which are challenging to formalize using mathematical equations or code. Although we can simplify their interactions to fit conventional methods, it often involves oversimplification and abstraction. LLM agents provide a more straightforward way to simulate these unstructured interactions that are challenging to formalize. We will elaborate on this point in the revised manuscript.
Done: lines 573 – 580.

**Technical corrections / typos**

- Line 199: "plausibility" is probably not the right word. "Desire for"?

- Line 327: Word missing (of?) in "investigation the large".

- Line 444: "conversions" should be "conversations".

- Line 472: Missing period.

➤ Thank you for pointing out these issues. We will correct these typos and rephrase them as suggested.
Done. Please see lines 263, 398, 517, 553.
* * *
**Reply on RC2**

**Major comments**

**Reproducibility**

- Please make all code and data used to produce these results available in a public repository with accompanying doi. In my view it is not enough to have code available "on request". This repository should have sufficient meta data and installation instructions to make your results fully reproducible on a range of operating systems.

➤ We appreciate your interest in the source code and data of our work. We will upload the code to a public repository with a DOI, ensuring it is accessible. We will also provide installation instructions. The code is implemented in Java and Python, both of which are cross-platform languages with APIs that should work across different operating systems. The repository will be open to public contributions, allowing others to adapt and extend the code, including testing and improving its compatibility with various systems. The author team will be glad to help with this.

Done. The data and code are now available at https://doi.org/10.5281/zenodo.14622334 and https://doi.org/10.5281/zenodo.14622039. The source code includes a README file that gives detailed instructions to install and use the model to enhance reproducibility.

- Lines 173-175; great that there is a CRAFTY emulator. However, I don't think this has been published. As such, to use it here I need you to provide details about its design and evaluation of how well it reproduces CRAFTY outputs as an SI. Please make all code and supporting data required to run it publicly available in a repository.

➢ Thank you for your comment. We are pleased to inform you that the CRAFTY emulator is publicly available on both GitHub and Zenodo, along with our other papers online. We will also upload a version of the source code that is coupled with the LLM agents for this paper to Zenodo with documentation.

The emulator has been designed to replicate the functions of the original CRAFTY model by the same team, and its behaviour has been evaluated to ensure alignment with the core dynamics of CRAFTY. As per your suggestion, we will add a description of the emulator's design and a comparison of its outputs with the original CRAFTY model in the SI of the revised manuscript.
A detailed description of the CRAFTY emulator and its output comparison with the main CRAFTY model can be seen in the Supplementary Information. Code and data links are given in the response above.

- Overall, I think a supplementary information expanding lines 179-191 to be really clear on what you did would be very helpful and make the work more reproducible. E.g. which RCPs / SSPs did you use? I *think* from Figure 5 that you are showing means across the scenarios, but this isn't totally clear. It would also be very interesting to see how much variance there is by scenario.

➢ Thank you for your suggestions. In this study, we used a single scenario, SSP1, to test the agents. The purpose of this paper is not scenario-specific; rather, it is intended as a proof of concept to explore the applications of LLM agents in land system modelling. By using a simplified setting, our aim is to observe the internal logic consistency and the contextually relevant behaviours of different agents rather than to test their performance across multiple RCP-SSP scenarios.

We acknowledge that testing the method across a range of scenarios could provide valuable insights, particularly if future research shifts toward a more empirical focus. In the revised manuscript, we will explicitly mention the scenario used in the experimental settings and clarify that the results are based on this single scenario.

We will expand the description in the Supplementary Information section to detail the coupled model works. We will also revise Fig. 5 and its caption to make it clear that the results are specific to one scenario.
Done. Please see Supplementary Information and manuscript with tracked changes from lines 228 to 256 for an improved description of the simulation processes. We also improved Fig. 3 by adding step numbers, corresponding better to its description in the text.

- Lines 126-152: do you have a complete set of prompts that you tried? Presumably this is possible if the dialogue is held in the agent memory (Line 139). I think this would be fascinating to see (as an SI) and would help with reproducibility.

➢ Thank you for your valuable suggestion. The dialogues were stored temporarily as a list of Python dictionaries in the program during runtime. We did not record how the prompts evolved. However,

we can provide the very beginning prompt (a draft actually) in the SI. While this does not capture the full evolution of the prompts, it might effectively illustrate the differences between the initial and final versions, which will be useful for understanding what changes have been made.
Done. Please see Supplementary Information.

- If not, then at lines 131-132 I think it would be good to more clearly state your criteria for assessing the LLM output. Perhaps in a table. IE were these very strict criteria (basic functionality) the only ones you used? Or did you also, subjectively, select for those that seemed to make sense? How many people evaluated the LLM outputs during the human in the loop stage to check you were applying any criteria consistently? I think a degree of subjectivity is inevitable, but worth being transparent about.

➢ The primary focus of our prompt refinement was on the functional aspects of the agents, such as ensuring that their outputs were correctly formatted to avoid disrupting the existing programmed model, aligned with the simulation's structural requirements, and operationally smooth within the coupled system. This process was more about fixing problems in existing prompts rather than creating entirely new strategies for the agents.

For example, in one scenario, the agent repeatedly tried to adjust the meat supply based on meat demand rather than aligning with the policy goal. To address this, we modified the prompts in different ways, such as adding notes to the prompts to emphasize the real target, attempting to clarify that meat demand was only reference information, not the target to pursue. However, this misunderstanding persisted. We eventually excluded meat demand information from the input entirely for some agents, such as Agent S1.1 and Agent S1.2.

You may notice examples of the required format for agents' outputs and notes such as, "Don't fake interaction with the policymaker if there is no interaction yet. " All such information was derived from the human-in-the-loop (HIL) process. Each note reflects a functional issue that was identified and addressed during the HIL process.

Given this functional focus, only one person was involved in the prompt refinement process. Having more people engaged would have been desirable to enhance consistency and reduce potential subjectivity, given that the process was time-intensive. However, the problematic outcomes were relatively straightforward to identify (e.g., ill-formatted outputs or repeated misunderstandings of context). A single person was able to carry out the necessary refinements effectively.

We will include this clarification in the revised manuscript and emphasize that while our approach was practical and focused on functionality, future research could benefit from broader collaboration in refining and evaluating prompts, particularly as more complex tasks are introduced.
In Supplementary Information, we have elaborated on what elements in the initial prompt were changed through the prompt refinement process.

**Scaling**

You note scaling issues in your discussion; however I think some further information and details would be helpful for the reader to gauge the extent of these challenges.

- It would be helpful to get a sense of run-time of one human-in-the-loop prompt session per institutional agent type (setup, LLM thinking time etc). E.g. Suppose we wanted to do sets of runs with differing levels of policy targets to assess consistency of answers. How feasible is this?

➢ Yes, this is highly feasible, particularly as more reliable and faster APIs are being developed. For example, GPT-4o, released in May this year, is significantly faster than GPT-4 used in this paper. Additionally, open-source LLMs accessed via platforms like Groq API offer specialized infrastructure that can speed up LLM inference, further enhancing feasibility.

During our experiments, the runtime for LLM agents' "thinking" was slower, but the major challenge was the occasional failure of API responses, which required re-sending requests. This made the runtime less predictable. Another time-consuming aspect was the iterative process of trial and error in refining prompts. While this approach allowed us to address specific issues, it did not guarantee consistent progress with every iteration.

It is worth mentioning that recent developments in frameworks, such as LangGraph's integration with Pydantic, help automate data validation and improve output structuring. While these tools are not yet perfect or fully mature, their ongoing development greatly facilitates the creation of AI agents and makes outputs adhere to the required format. These technologies could significantly reduce the time and effort required for prompt refinement and data format validation in future research.

- Similarly, perhaps I missed this, but if you repeat one of your policy scenarios multiple times, how much do the resulting outputs diverge? Is this computationally prohibitive?

➢ The results generated by LLMs can be reproducible under specific conditions. Reproducibility depends on factors such as setting a fixed random seed, if the system or API allows, using the exact same model version and configuration (e.g., temperature, max tokens), ensuring that input text remains identical.

With the same settings and running the API on the same device, the model can reproduce the same results in principle. This is what we have observed during our experiments. However, exact reproducibility might not always be feasible because users cannot control when or how LLM providers might update the underlying model. These updates can result in subtle differences in output even when the same inputs and parameters are used.

This is an important caveat for reproducibility, which we will highlight in the revised manuscript. We also suggest that our focus on evaluating the internal logical consistency and contextual relevance of the LLM agents' outputs is expected to be relatively robust to these changes.

- Further, if you have stochasticity in the underlying model, how much can this lead to unpredictable policy pathways? You provided historical data to the LLM – I take it this was observations rather than historical CRAFTY runs? Did you then spin-up the CRAFTY AFT distribution to match the observations? Otherwise might our initial LLM choices be sensitive to the initial conditions?

➢ A good point; stochasticity exists in both the LLMs and the CRAFTY model, although there is no spin-up and the model starts from the same set of conditions each time. As above, we will discuss the existence and influence of stochasticity in the revision. It's worth noting though that the data received by the LLM agents are updated periodically during the simulation as CRAFTY runs. This ensures the agents are informed by dynamic, real-time simulation outputs rather than relying solely on static, pre-observed (historical) data. We will revise this phrasing in the paper to eliminate any potential confusion.

While both models are able to produce understandable behavioural patterns, which is why they are useful and can be coupled meaningfully, stochasticity can be very important. In the case of LLMs, a straightforward way to increase unpredictability is by adjusting the temperature parameter. Higher temperatures make outputs more diverse and therefore more unpredictable across runs. However, in our experiments, the temperature was set to 0 by default, which is intended to ensure consistent outputs across runs.

Even if some unpredictability exists, it might not impact the goals of this research. Our primary focus is on the believability, contextual awareness, and logical consistency of the LLM outputs, as well as their potential to mimic human decision-makers with understandable behaviours. If unpredictability enhances the diversity of the LLMs without compromising the quality of their outputs, they remain valuable and worthy of further study.
We added lines 666 to 687 to discuss reproducibility.

**Generalisability**

You observe that stakeholder disagreement & subsequent contested policy spaces lead generally to slower decision-making. This is an important and fundamental insight, with some grounding in the literature. Some questions and comments below on how universal / generalisable such a finding may / may not be.

– Lines 308-312

The setup of agent Q overall seems good and appropriate. I think it is worth being careful to remind readers that you are explicitly mimicking policymaking processes in a European context with broadly democratic norms and systems. The text here seems to discuss multi- stakeholder policymaking in abstract terms, but the setup of the multi-stakeholder network would presumably have to vary substantially in other policymaking systems. For example, in more authoritarian government systems, we may have "industrialisation from above" with very rapid changes, or one group of stakeholders' rights and views being cut out of decision-making.

➤ Thank you for your insightful comment. We agree with your observation and will clarify in the text that the setup of Agent Q reflects a political system modelled on broadly European Union (EU)-like democratic norms and systems, rather than an authoritarian framework. We acknowledge that the dynamics of multi-stakeholder policymaking would differ significantly in other political contexts, such as in authoritarian systems where rapid "industrialization from above" or exclusion of certain stakeholder groups may dominate the decision-making process. We will ensure this distinction is explicitly stated in the paper to avoid potential misunderstanding.
Done: lines 509 – 511.

– Lines 407-414

Here and elsewhere you state that slow and or incremental policy changes are more realistic /more in line with expectations than the optimisation algorithm. A few more references to support this would be good, particularly to clarify whether this is primarily a feature of western democratic systems or a more general phenomenon.

That said, let us assume that incrementalism is a broadly realistic simulated policymaking approach. I wonder if, in a subsequent paper, one could demonstrate this empirically? E.g. could we take countries' stated climate targets, and review concrete progress / policy implementation towards them vs what an economically-optimal trajectory towards achieving
them might look like. If it could be clearly demonstrated that these simulated policy responses are closer to real-world choices than optimisation-based modelling that would be a tremendously important finding, I think. Not only to evaluate your model, but also for wider consideration of institutional constraints on rates of environmental land use change.

➤ We agree that incrementalism, as a feature of policymaking, could benefit from stronger contextual support in the paper. Incremental policy change has been well-documented as a characteristic of western democratic systems, particularly in literature on policy sciences. We will incorporate additional references to highlight these points.

Your suggestion to empirically validate incremental policymaking approaches by comparing real-world policy trajectories to simulated and optimization-based trajectories is highly valuable. We see significant potential for future research in this direction. It can be envisioned that computational expense might be a critical challenge in applying optimization algorithms to policymaking and land-use modelling. When institutional models or land-use models are highly complex or involve large parameter spaces, optimization can become infeasible due to excessive computational requirements. Moreover, if the optimization process is not sufficiently robust, it may underperform heuristic or rule-based decision-making approaches in both accuracy and speed, given the time required for computation.

In this paper, the optimization algorithm operates within a limited action space, which enables it to work effectively. However, this is a simplified scenario designed specifically for this research's conceptual focus. Expanding the optimization to a broader or more realistic problem space would introduce significant complexity. Reaching a balance between problem complexity and computational feasibility is critical, and defining the optimization problem well is a prerequisite to ensure it is neither too complicated to yield effective solutions nor too simplistic to preserve the essence of empirical policymaking processes.

We found that Discussion might be the most suitable section to include more text and references about incrementalism. Please see lines 527 to 534.

**Minor comments**

– Line 67: I think it would be worth flagging here that the majority of this training text is in English and noting how this may culturally skew the "thinking" of the LLM. This sets up the issues around generalisability.

➢ Thank you for pointing this out. It is true that the richness and diversity of training text in different languages significantly affect the performance of large language models (LLMs). We will note in the paper that English is one of the richest and most extensively represented resources in LLM training datasets. This can lead to a cultural and linguistic skew in the "thinking" of the LLM, with potentially different outputs and text generation patterns. This highlights an important limitation and sets up the issue of generalisability, which we will explicitly discuss in the revised manuscript.

Done: lines 632 – 634.

– Lines 67-74; I'm not going to argue that we need to move beyond the paradigm of economic rationality. That said, some citations here would be good -> can we be explicit about whyeconomic rationality may not produce nuanced representations of human decision making?

➢ We agree that it is important to explicitly address why economic rationality may fall short in representing the nuances of human decision-making. While economic rationality assumes that individuals always act in ways that maximize utility based on stable preferences and perfect information, human decision-making is often influenced by factors such as bounded rationality, cognitive biases, emotional responses, and social or cultural norms. These factors lead to behaviours that may deviate from purely economically rational decisions.

We will incorporate citations to support this perspective. Foundational works such as Simon's *bounded rationality* (1997) and Kahneman and Tversky's *prospect theory* (1979) will be included to contextualize this limitation of economic rationality. This will help clarify why alternative approaches, such as those explored in our model, are valuable for capturing the complexity and nuance of human decision-making.

After careful consideration, we found inserting an explanation here could affect the flow of text. We must acknowledge that the inclusion of economic rationality slightly deviated from the main topic here. So, we chose to delete the original text and cite a reference to indicate LLMs' capability in modelling multifaced and nuanced human behaviours. Please see lines 74 – 77.

– Lines 87-95; this is good, but I want several citations here to point users not familiar with the LLM literature in the right direction. If I wanted to replicate your method, I would want first to consult the underlying methodological literature on this topic.

➤ Sure, we will add relevant literature in the text for the reader to reference.

Done. Please see lines 117 to 124.

– Similarly, lines 95-119; it isn't clear to me as a non-expert how far this is methodological innovation or a very common approach to prompt development. I would like much more reference to the underlying literature. Please also add these to Figure 1 as 1,2,3, etc with refs in a key.

➤ The prompt development framework proposed in this paper provides a systematic process for developing prompts specifically for LLM agents that are coupled with existing programmed systems. While some of the techniques may align with common practices in prompt engineering, our contribution lies in presenting a streamlined and structured process tailored to this particular task. This framework aims to assist future researchers in accelerating their prompt development by providing clear steps.

At the time this paper was composed, there were numerous papers and technical documents discussing prompting techniques in general. However, resources specifically addressing the development of prompts for LLMs integrated with external programmed systems were very limited, if not nonexistent. To our knowledge, this framework represents a novel contribution. It is not entirely new in every individual element, but rather it synthesizes key steps derived from the engineering practices developed in this research.

This framework is more of a practical engineering summary than a theoretical model supported by established evidence. Given the rapid developments in this field, we will review recent literature to identify relevant works that can be incorporated as references for the paper.

Our framework is aimed to develop prompts to instruct LLM agents coupled with existing dynamic simulations and is rooted in our practical experiences. While it is possible that other researchers with similar needs may develop comparable frameworks, our literature search did not reveal existing frameworks identical or similar to the one we propose in this paper. For those interested in a broader understanding of recent prompting techniques, we recommend referring to the added lines 117 to 124 for broader references.

In lines 125 to 132, we highlighted the distinctiveness and compatibility of our framework with existing prompting techniques.

In lines 152 to 154, we enhanced the explanation of fake loop testing by comparing it with the "mocking" technique, a widely used method to test programs in object-oriented programming.

– Figure 2: This is good, but please can it be formatted so that I understand better where it begins and ends & which steps follow which? Either by numbering stages or having it more clearly as a top-to-bottom process flow? Figure 3 is much better in this regard.

➤ Of course, we will improve Figure 2 to make the processes clearer.

Done. Please see Figure 2 and its revised caption. In addition to rearranging the layout of the diagram, the new figure omitted the arrow indicating information flow rather than a procedure

transition from "proposal" and "Prompt Update" to improve clarity.

– Figure 5: Please point the reader to Table 1 for definition of your agent types in the caption.

➤ Very constructive suggestion. Thank you. We will mention Table 1 in the figure caption to better guide readers.

Done. Please see line 337.

**References**

Simon, H. A. (1997). Models of bounded rationality: Empirically grounded economic reason. MIT press.

Kahneman, D., & Tversky, A. (1979). Prospect Theory: An Analysis of Decision under Risk. Econometrica, 47(2), 263–291. https://doi.org/10.2307/1914185.